# Recent Progress in Perovskite Tandem Solar Cells

**DOI:** 10.3390/nano13121886

**Published:** 2023-06-19

**Authors:** Steponas Ašmontas, Muhammad Mujahid

**Affiliations:** Center for Physical Sciences and Technology, Saulėtekio Ave. 3, LT-10257 Vilnius, Lithuania; muhammad.mujahid@ftmc.lt

**Keywords:** solar cells, perovskite, tandem configuration, bandgap tune ability, stability

## Abstract

Tandem solar cells are widely considered the industry’s next step in photovoltaics because of their excellent power conversion efficiency. Since halide perovskite absorber material was developed, it has been feasible to develop tandem solar cells that are more efficient. The European Solar Test Installation has verified a 32.5% efficiency for perovskite/silicon tandem solar cells. There has been an increase in the perovskite/Si tandem devices’ power conversion efficiency, but it is still not as high as it might be. Their instability and difficulties in large-area realization are significant challenges in commercialization. In the first part of this overview, we set the stage by discussing the background of tandem solar cells and their development over time. Subsequently, a concise summary of recent advancements in perovskite tandem solar cells utilizing various device topologies is presented. In addition, we explore the many possible configurations of tandem module technology: the present work addresses the characteristics and efficacy of 2T monolithic and mechanically stacked four-terminal devices. Next, we explore ways to boost perovskite tandem solar cells’ power conversion efficiencies. Recent advancements in the efficiency of tandem cells are described, along with the limitations that are still restricting their efficiency. Stability is also a significant hurdle in commercializing such devices, so we proposed eliminating ion migration as a cornerstone strategy for solving intrinsic instability problems.

## 1. Introduction

Any viable approach to lowering CO_2_ emissions and preventing another rise in average temperature would have to include alternative sources at its cornerstone, since fossil fuels constitute approximately 80% of the world’s energy consumption [1]. As a result, solar photovoltaic (PV) technologies have garnered enormous levels of societal attention: market forces are fast growing their investments in PV, despite continued attempts by research organizations to examine the principles of converting solar energy to electricity to push efficiency restrictions. Regarding producing electricity, PV is leading the pack regarding growth rate. There was a dramatic drop in manufacturing costs, installation, and maintenance of solar systems in the last decade. The restricted power conversion efficiency (PCE) and the project’s capacity, which remains significantly elevated compared to the expenses associated with non-renewable energy sources, suggest that this technology may not be the most widely used primary grid energy source. Even so, this technology could become a source of energy in the coming decades [2].

Silicon crystalline material single-junction solar cells (SCs) dominate the market, used in producing commercial solar modules because of their low production prices and the outstanding dependability of their materials and manufacturing technologies [3]. Crystalline silicon solar panels, with a maximum PCE of 26%, have dominated the PV industry because of their efficiency and reliability. These devices’ efficiency is close to the theoretical maximum limit of 33.3%, according to Shockley–Queisser’s (SQ) detailed balance model for ideal p–n junction [4]. A key limiting factor not accounted for in the SQ model is the Auger recombination of free carriers that occurs under illumination. Taking this into account for silicon, the efficiency limit for monocrystalline Si SC with an optimized thickness (110 μm) was calculated to be 29.4% [5,6]. The two primary problems limiting single-junction solar cell performance are thermalization losses and the non-absorption of low-energy photons that fall under the bandgap. Several strategies have been implemented to effectively meet the Shockley–Queisser (SQ) limit and collect the maximum number of photons possible. The concept of intermediate-band solar cells has been proposed to achieve a conversion efficiency of 63% [7]. Quantum dots have been considered one of few materials systems to form intermediate bands for intermediate-band solar cells [8]. The process of carrier multiplication, in which a single photon generated two (or more) electro-hole pairs can enhance the photocurrent of SCs [9]. Metallization at the nanoscale is also a method that can be used to improve SCs’ efficiency [10]. Incorporating absorber materials into a multi-junction arrangement is one of the approaches that can be taken. These materials should have different band gaps. The amalgamation of a material possessing a significant bandgap and one with a low bandgap on top of one another is referred to as a tandem configuration. Since the SQ criteria constrain single-junction solar cells, integrating subcells with varying bandgaps is the quickest and easiest route to surpassing the SQ limit of single p–n junction [11]. Studies have revealed that compound semiconductors, such as InP and GaAs, are used during the construction of III–V multi-junction solar cells, and they are exceptionally efficient. The six-junction tandem solar cell has established an impressive 39.2% one-sun efficiency, constructed using III–V compound semiconductors [12]. Nevertheless, the significant production of such materials is problematic because of the high costs and the complicated manufacturing procedures.

The advancement of tandem devices utilizing III–V semiconductor materials and silicon has been considerably influenced by the lattice distortion and thermal expansion coefficient discrepancies between the two materials. In addition to the high capital and operating costs, tandem devices struggled with these issues [13]. Utilizing the advantages of perovskite materials—known for their direct bandgap, high absorption coefficient, and superior charge transport properties—researchers have been designing and optimizing tandem solar cells. These perovskite tandem solar cells typically consist of a perovskite top cell paired with a bottom cell, often composed of silicon or another perovskite variant. This configuration broadens the solar spectrum coverage, thereby amplifying overall efficiency. Moreover, advancements have been made in improving the stability of these cells, with notable progress in encapsulation techniques and tweaks to perovskite composition that significantly reduce degradation over time. Despite these strides, challenges persist in areas such as large-scale manufacturing, consistency in the fabrication of thin films, and long-term stability, among others. However, the potential of perovskite tandem solar cells continues to drive substantial research and commercial interest. The basic idea of the article is provided in Figure 1.

Recent research has shown that greater PCEs can be obtained using organic–inorganic hybrid halide perovskite materials (CH_3_NH_3_PbI_3_) instead of Si single-junction cells with advanced PV technology. For this reason, hybrid metallic halide perovskite materials, which are both organic and inorganic, share the benefits of both types of compounds. Because their performances increase higher than silicon-based PV technology, the newly developed perovskite solar cell (PSC) has been recognized as something that might drastically change the PV market. PSCs have captured significant interest in the solar world due to the higher efficiency, minimal cost, process compatibility, and adjustability of their bandgap. With their widely-accepted bandgap of 1.1 eV, high open-circuit voltage (V_OC_) of up to 750 mV, cheap manufacturing due to their supremacy in the industry, and exceptionally high efficiency, crystalline silicon solar cells are almost perfect for use in these tandem cells [5,14]. Locating a suitable wide-bandgap (WB) component is far more complicated. Due to their enhanced efficiency and adjustable bandgap, III–V solar cells have been frequently reported upon. Recent research has shown a mechanically stacked four-terminal tandem cell that has a demonstrated efficiency of 32% [15].

Nevertheless, the widespread use of III–V solar cells’ expensive manufacturing processes prevents them from being used for practical uses. The advancement of perovskite-based PV technology provided a viable solution to the need for inexpensive high-efficiency solar panels [16]. The potential PCE produced using tandem solar cells (TSC) increases with the number of light absorbers exhibiting diverse bandgaps. One example is the current PCEs of 32.9%, 37.9%, and 39.2% for III–V multi-junction solar cells with a non-concentrator and 2, 3, or 6 connections, respectively [17]. The high price of tandem cells may be reduced using metal halide perovskite solar cell technology. The excellent absorption coefficient [18], exceptional bandgap tunability [19], the extended range of charge carrier diffusion [20], and minimal binding energy for excitons [21] are only a few of the optoelectronic qualities exhibited by metal halide perovskites that make them ideal for producing high PCEs in solar cells. The relatively lower crystallization temperatures and solution processability contribute to the low fabrication costs, making this material not only economically viable but also marketable [22] With 25.2% PCE, solution-processed single-junction metal halide PSCs outperform standard c–Si solar cells. The compositional engineering of the A, B, and X sites allows the bandgap of ABX_3_ metal halide perovskites to be tuned, making them suitable for deployment as an upper cell on any bottom cell. At this point, A indicates a monovalent cation, while B represents a divalent metal cation and X denotes a monovalent halogen anion. Furthermore, the bottom cell is much less likely to be damaged because of the low-temperature fabrication procedure—various solution process deposition methods, including spin coating [23] or slot-die coating [24]. In addition, vacuum-based methods, such as thermal evaporation or chemical vapor deposition [25], are reported as manufacturing protocols for PSC.

## 2. Halide Perovskite Materials for Solar Cells

Inorganic–organic halide solar cell researchers have become very interested in PSCs due to a striking rise in device efficiency from 3.8% [21] to 25.8% [22] since 2009. Considering an all-time high efficiency of around 26.7%, silicon PV systems hold most of the market share; perovskite has attracted much interest [26]. Given that certain materials have the potential to be employed in both photovoltaic PSCs and organic solar cells, this small efficiency difference has recently caught the attention of researchers, particularly those with expertise in dye-sensitized solar cells (DSSCs) or organic solar cells. The DSSC device architecture is where PSCs display advantages over silicon-based devices, as seen in [27], which require labor-intensive and expensive high-vacuum deposition procedures. Based on the accounts of efficacious cell production on flexible substrates, further promise exists in that the extensive roll-to-roll fabrication of PSCs could be implemented in various industrial sectors [28,29]. In 1839, German mineralogist Gustav Rose discovered a calcium titanate crystal structure. The nomenclature “perovskite” was assigned to this structure in honor of Lev Perovski, the Russian mineralogist who initially unveiled it; since then, the name “perovskites” has been used to describe all materials that share calcium titanate’s crystal structure. The general formula for the perovskite light absorption layer is ABX_3_, where A is an organic cation (CH_3_NH_3_^+^), B is a metal cation (Pb^2+^), and X is a halide anion (I^−^).

### 2.1. Crystal Structure

In Figure 2a–c, we can observe that molecular structure of the organic halide perovskite is of the ABX_3_ type, where A and B are cations (with A being bigger than B) and X is the anion. The A cation (methylammonium, CH_3_NH_3_^+^, MA^+^, or formamidinium, CH_3_ (NH_2_)_2_^+^, FA^+^) in a perovskite’s unit cell is surrounded by 12 X anions (Cl^−^, Br^−^, or I^−^, or a coexistence of multiple halogens) to form a cuboctahedron. The octahedral position of X is occupied by the B cation (Pb_2_^+^, Sn_2_^+^, etc.) The octahedra of the B cation and X anion are connected to create a stable three-dimensional network structure [30,31,32]. For CH_3_NH_3_PbX_3_, as the size of the halide grows from X = Cl to Br, the unit cell parameter grows from 5.68 to 5.92 and then to 6.27 Å. Mixing halides makes it easy to adjust the cubic phase’s lattice characteristics; for example, CH_3_NH_3_PbBr_2.3_l_0.7_, CH_3_NH_3_PbBr_2.07_I_0.93_, and CH_3_NH_3_PbBr_0.45_I_2.55_ all displayed a = 5.98 Å, 6.03 Å, and 6.25 Å, respectively. The cubic perovskite structure of CH_3_NH_3_SnBr_x_l_3−x_ (x = 0–3) crystallized unit cell parameters are reported as follows: a = 5.89 Å (x = 3), a = 6.01 Å (x = 2), and a = 6.24 Å (x = 0). Some Sn-based perovskite compounds showed conducting capabilities in contrast to Pb-based perovskite materials [33,34].

A dimensionless value termed the Goldschmidt tolerance factor (t), which measures the crystal’s stability and its structure’s deformability, is used to predict the formability of different kinds of perovskites.
(1)$$t=\frac{r_{A}+r_{x}}{\sqrt{2}\left(r_{B}+r_{x}\right)}$$
where the radius of cation A is r_A_, the radius of cation B is r_B_, and the radius of anion X is r_X_. For perovskite composed of a transition metal cation and an oxide anion, t = 1 is projected to result in a cubic structure, while t < 1 is predicted to result in octahedral deformation. Additionally, at t < 1, symmetry is lost, which impacts the electrical characteristics. Perovskite formability for alkali metal halides is anticipated to be 0.813 < t < 1.107 [35,36]. Cubic structures have tolerance factors between 0.89 and 1, with values greater than 0.89; due to the instability of B–X bonding in 3D, a transition to a 2D layer structure may occur. In contrast, a tetragonal (β phase) or orthorhombic (γ phase) structure may form if the tolerance factor is less than 0.89 [37]. The octahedral factor (µ), provided by Equation (2), is another component for perovskite formation.
(2)$$µ=\frac{r_{B}}{r_{x}}$$

Perovskite stability and distortion can be assessed through its utilization. The perovskite can have an octahedral factor between 0.45 and 0.89 without losing stability. Methylammonium lead trihalide (MAPbI_3_, where X is the halide, could be Cl, Br, or I) is the most widely utilized absorber substance for PSC. As the size of the halide atom grows from Cl to Br to I, the unit cell characteristics rise from 5.68 to 5.92 to 6.27 Å. However, the increased size and aspherical shape of methylammonium (MA) cause the network to distort, which causes a phase change and a drop in temperature. The orthorhombic structure is present for T < 160 K, the tetragonal structure for T > 162.2 K to T < 327.4 K, and the cubic structure for T > 327.4 K. The bandgap of methylammonium lead halide typically ranges from 1.5 to 2.3 eV; MAPbI_3_ has a direct bandgap of roughly 1.55 eV, whereas MAPbBr_3_ has a relative WB of 2.3 eV for 600 nm absorption started. FAPbI_3_ (where FA is formamidinium) exhibits a NB of 1.48 eV as the absorber layer, indicating greater current extraction. However, this material has demonstrated lesser stability. The type of halogen atom employed also affects the structural characteristics of methylammonium lead halide. Lead halide perovskites have an octahedral crystal structure, as illustrated in Figure 3a. The B cation (typically Pb but sometimes Sn) is octahedrally coupled to six halide ions. There is a shared corner between these octahedra, and the A cation lies between them [38,39]. The A cation of lead halide perovskites is either an organic molecule (methylammonium, MA, CH_3_NH_3_^+^) or an inorganic cation (usually Cs^+^), leading to the further categorization of lead halide perovskites as either organic–inorganic (hybrid) or all-inorganic. One can manipulate perovskites’ optical and electrical characteristics by changing the proportions of formed halide ions and, to a lesser extent, the cations [40]. Like traditional metal chalcogenide semiconductors, perovskites’ optical characteristics and object tuning can be achieved by adjusting their dimensions and size. Perovskites exhibit a pronounced inclination towards the creation of stratified two-dimensional (2D) and quasi-two-dimensional architectures (as depicted in Figure 3c), even though their dimensional range can be altered from three-dimensional to zero-dimensional by manipulating the synthetic parameters employed (Figure 3b) [41,41,42,42,43]. Numerous recent investigations have demonstrated the excellent photoluminescence quantum yields of colloidal perovskite nanocrystals. Furthermore, when their dimensional changes are reduced from 3D to 2D, they exhibit significant quantum confinement effects, allowing for the further customization of optical characteristics [44,45]. A good number of applications, such as solar cells and lasers, LEDs, and PV, have demonstrated the immense potential of perovskite.

### 2.2. Electronic Structure

Here, by considering the perovskite material’s electronic characteristics to understand its layered perovskite structure. Density functional theory (DFT) computer simulations were used in several theoretical investigations on the density of states (DOS) of perovskites, correlating the composition of the bands with the DOS. We may learn more about where those remarkable optoelectronic capabilities of perovskite materials come from and how they can be optimized with the help of the DFT calculations performed on metal halide perovskites. The valence band results from the overlap between the M-site cation’s ns orbitals and the X-site anions’ np orbitals. In contrast, the conduction band is formed by combining the np orbitals of the M cation and the X anions [18,46]. It is well established that there is an antibonding feature between the valence band maximum (VBM) and the conduction band minimum (CBM) because of the orbital overlap among the filled ns of M and np of X [47,48]. As depicted in Figure 4, the ionic nature of bonds expands, and E_g_ increases when the composition shifts from I to Cl because the energy difference between ns and np halogen atomic orbitals widens.

M metals have a less noticeable impact on the band structure. The final band structure is influenced by phenomena such as ns^2^ lone pairs, the relativistic stabilization of the 6s^2^ level in Pb^2+^, and the composition of M–X bonds (degree of covalence). Considering all these characteristics, the E_g_ of these materials can be precisely tuned by altering their compositions or making alloys. In addition, the absence of electron–phonon coupling in perovskites’ straight bandgap makes electronic transitions more likely. Since the electron’s quantized wave motion in a periodic crystalline lattice is connected to its crystal momentum (vector k), the VBM and CBM in direct bandgap materials are coplanar about this axis. Yin et al. computed the band structure and DOS of the cubic phase MAPbI_3_ using DFT–PBE computation, as illustrated in Figure 5. The band structures of the tetragonal and orthorhombic phases of MAPbI_3_ are very close to those of the cubic phase, according to a subsequent investigation by the same group. According to the experimental findings, in the tetragonal and cubic phases, the bandgaps are expected to be 1.55 and 1.57 eV, respectively, and the UV-vis measurement of FASnI_3_ is illustrated in Figure 5c [49,50]. Since the VBM and CBM coincide at the same point of the Brillouin zone, cubic MAPbI_3_ is a direct bandgap semiconductor (Figure 5a). The two MAPbI_3_ phases, tetragonal and orthorhombic, are direct bandgap semiconductors exhibiting values that are nearly close to one another. The comparison between the experimental UV-vis spectrum of MAPbI_3_ (red line) and the SOC–GW calculated spectrum (blue line) is depicted in Figure 5d [49,51]. Figure 5b displays the effects of MA, Pb, and I on the DOS of MAPbI_3_. The Pb *p* orbital dominates the CBM, while I *p* states, through a minor addition from Pb *s* states, dominate the VBM. Given its position well below the VBM, the MA cation’s partial DOS does not make any direct electronic contributions to the CBM or VBM. MAPbI_3_’s remarkable electrical properties can be traced back to a lone pair of s orbital electrons in the Pb cation. While the outer s orbitals of most metal cations are empty, Pb possesses an occupied 6s orbital just under the valence band [18,52].

### 2.3. Structure of Perovskite Solar Cells

Efficient PSCs could be the most fantastic solution for commercial solar technology manufacturing. PSC products are likewise separated into two categories for a good reason, namely normal and inverted, as shown in Figure 6 [53]. The hole transport layer (HTL) or the electron transport layer (ETL) should be placed first; there are two distinct methods for fabricating PSCs. Semiconducting p-type polymers create an inverted structure, such as PEDOT:PSS, while the standard structure uses n-type semiconductors, such as TiO_2_. The first ETL employing titanium oxide (TiO_2_) was built in 2009, and it was the predecessor of the current structure. A PIN-type product based on the organic HTL poly(3,4-ethylene dioxythiophene)-co-polystyrene sulfate (PEDOT:PSS) was released four years later. These two architectures currently have highest PCEs reported [27,54,55,56,57]. By employing growth methods, the fabrication temperature for n–i–p structured flexible PSCs could be kept to a minimum. However, an n–i–p structure device requires an excessive temperature method to develop a compact TiO_2_ coating. Plastic is not a suitable surface for this [53]. The mesoporous TIO_2_ film was initially employed as a scaffold for the initial PSCs fabricated from dye-sensitive solar cells (DSCs); as a result of sintering nanoparticles (NP), a porous TiO_2_ layer developed, with the self-assembled perovskite absorber providing the filler. A typical arrangement of such PSCs is shown in Figure 6a.

It is easier to transfer electrons between the FTO electrode and the perovskite absorber, and a perovskite coating was formed on mesoporous TiO_2_ [58]. Perovskite NPs work to progressively strip the photoexcited pigment of its electrons, enabling it to serve mainly as a light absorber in place of the molecular sensitizer it formerly was. However, the finding that organic–inorganic halide perovskite could independently conduct electrons and holes encouraged the creation of future devices. The mesoscopic variant’s perovskite capping layer extends well beyond the nanostructure thanks to the semiconductor oxide scaffold’s thoroughly infiltrated pores. The planar structure was researched to facilitate fabrication. Planar metal oxide ETL fabrication typically occurs at temperatures below 200 °C without impairing the perovskite devices’ functionality. The standard structure is thus receiving a great deal of attention for further research [59,60]. Such a lower temperatures solution technique might result in the fabrication of the ETL, so the p–i–n-type arrangement has gained even greater notoriety.

Along with the electron-transporting material (ETM) layer, a hole-transporting material (HTM) layer was also applied; perovskites were suited for hole transport. This has been a major component in manufacturing p–i–n-structured solar cells. HTLs are well-suited for adaptable perovskite optoelectrical devices, since their production often does not need a high-temperature annealing procedure [61,62]. HTM is also an air- and water-resistant alternative for PSCs. To prevent leakage, the objective of producing planar p–i–n on a uniform transparent conductive oxides (TCO) electrode is to manufacture a perovskite layer without pinholes by utilizing a one-step spin-coating procedure. Results from cells produced from a 1:1 molar PbI_2_/CH_3_NH_3_I solution are inferior to those from the 1:3 molar PbCl_2_/CH_3_NH_3_I solutions. The annealed perovskite film still exhibited pores and poor crystallinity since no scaffold had been used at 100 °C [63,64]. These findings prove that the perovskite layer’s appearance and crystallinity are critical to the device’s performance.

## 3. Perovskite Materials for Tandem Solar Cells

In perovskite-based tandem devices, perovskites with compositionally maintained bandgaps are frequently used. These perovskites have a wide-bandgap of >1.55 eV and a narrow-bandgap of less than 1.35 eV. Hybrid perovskites exhibit a direct bandgap, possess a high degree of optical absorption, demonstrate balanced and small active area for both electrons and holes, exhibit a high degree of tolerance for defects, display extended carrier lifetimes and diffusion lengths, feature low exciton binding energies, and exhibit completely innocuous grain boundaries [20,65,66,67]. The distinct electronic structure of materials composed of perovskite is the reason for their exceptional optoelectronic properties, contributing to the excessive V_oc_ and efficiency of PSCs. The combinations of A-site cations (such as methylammonium (MA), formamidinium (FA), caesium, and rubidium), B-site cations (such as lead and tin), and X-site anions could be used to accurately control the bandgap of the perovskite absorber to a certain level (such as iodide, bromine, and chloride). Subcells that perfectly fill the bandgap may be fabricated using this bandgap-engineering method [40,68,69,70,71]. Hybrid perovskite components are abundant and inexpensive. In addition, it may be manufactured on flexible substrates using a solution technique at lower temperatures, allowing for cost-effective large-scale manufacturing [72].

### 3.1. Wide-Bandgap for Perovskite Top Cell

The typical symbol for the crystal structure of metal halide perovskite (MHP) is ABX_3_. The impact of A-site cations on the bandgap is attributed to their ability to induce lattice distortion in MHP, which impacts the bond length and angle of the B–X sites, ultimately influencing the bandgap. Since B-site cations connect angles within the BX_6_ octahedron, the bandgap decreases with increasing angle. The band gap was shrunk by increasing the X-site halogen anions’ ionic radius and decreasing the B–X bonds’ valence level [17]. The compositional engineering of methylammonium (MA), formamidinium (FA), caesium (Cs), and rubidium (Rb) as A-site cations, lead (Pb), tin (Sn), and germanium (Ge) as B site cations, and iodine (I), bromine (Br), and chlorine (Cl) as the X site anion allows for the fine-tuning of the bandgap in ABX_3_ [40,73]. Figure 7 shows that the bandgap of MHPs can be adjusted from the infrared area (~1.1 eV) to the ultraviolet region (~3.0 eV). In combination with NB cells, this shows that MHPs provide high flexibility. Due to the processing temperature below 150 °C, MHPs allow monolithic configurations to be fabricated without harming the bottom cell. Shallow trap sites, long carrier diffusion distances, excessive dielectric constants, fewer exciton binding energies, and significant absorption coefficients are further benefits [72,74,75,76]. MHPs have been recommended as the top cell’s light-absorbing layer in various TSCs. WB perovskites are extensively employed, surpassing other materials in terms of usage in TSCs; MAPbI_3_ has attained extraordinarily high efficiencies. This demonstrates the versatility of MHPs when used with NB cells, starting with a low 13.4% efficiency using a MAPbI_3_/silicon heterojunction (SHJ) 4T tandem structure, MAPbI_3_-based tandem devices grew to 27.0% efficiency after extensive tuning, outperforming even the most efficient crystalline silicon solar cells (which achieve 26.7% efficiency) [5,77,78]. It is conceivable that the use of MAPbI_3_ perovskite affected the development of TSCs based on perovskites. However, when subjected to atmospheric factors, including oxygen, heat, light, and moisture, MAPbI_3_ undergoes a chain reaction that includes chemical reactions, phase transitions, phase segregation, and other forms of degradation. The bandgap in MAPbBr_3−x_I_x_ could be controlled chemically within the range of 1.55 to 2.3 eV, with x changing from iodide to bromide. Top subcells in a tandem arrangement can achieve 1.70–1.85 eV bandgaps. However, it has been established that when the bromide concentration is too high, light instability results in photo-induced halide segregation in MAPbBr_3−x_I_x_, which lowers the achievable voltage and reduces the functionality and dependability of mixed-halide perovskite devices [40,79,80].

### 3.2. Narrow-Bandgap Perovskites for Bottom Cells

High efficiency can be achieved with all-perovskite tandems while retaining the advantages of low price, low-temperature manufacturing, and the potential for both subcells in the structure to have light, flexible form factors. Substituting tin Sn for Pb is a feasible method for decreasing the perovskite bandgap. The optical bandgap in Sn-based perovskites is narrower than in Pb-based perovskites, but the isoelectronic configuration is the same [81]. It has been found that SnPb-alloyed perovskites are the sole method capable of achieving an E_g_ as low as 1.2 eV, which maximizes the solar spectrum’s use in conjunction with wide-E_g_ perovskite resulting in exceptionally efficient all-perovskite tandem cells. At a 60% Sn ratio, the bandgap in perovskite materials is the narrowest due to the bowing effect of the bandgap [82].

Conversely, perovskites containing Sn exhibit characteristics of semiconductors, including a notable intrinsic carrier density resulting from the spontaneous oxidation of Sn_2_^+^ ions and a short carrier lifetime caused by a significant trap density [83]. Snaith et al. initially reported on fabricating PSCs utilizing methylammonium tin iodide (CH_3_NH_3_SnI_3_) and deposited it on glass substrates in 2014. The bandgap of CH_3_NH_3_SnI_3_ has been determined to be 1.23 eV through absorption measurements [83]. The production energy of Sn^2+^ vacancy in Sn-based perovskites is low, and these materials have a propensity for oxidation to Sn^4+^ and are limited in their ability to exhibit n-type conductivity rather than the more common p-type [84]. The compound FA_0.6_MA_0.4_Sn_0.6_Pb_0.4_I_3_, wherein MA denotes methylammonium and FA denotes formamidinium, has demonstrated superior performance as a single-junction cell featuring a narrow-gap perovskite. With a bandgap of 1.25 eV, it has a PCE of 17.8% and a 4T tandem of 21% efficiency [85]. There have been reports of perovskites with even smaller bandgaps (MASn_0.8_Pb_0.2_I_3_ with 1.19 eV). They have not yet, however, created solar cells with a PCE of greater than 10% [73]. To obtain the ideal bottom cell bandgap for a 2T tandem, using perovskites with smaller bandgaps is preferable. It also makes it possible to employ more stable, smaller bandgap perovskites for the top cell, as mentioned above. According to Freeman et al., the optical bandgap of MASn_x_Pb_1−x_I_3_ perovskites is approximately ~1.25 eV, despite the absorption starting at 1060 nm, indicating a minimum bandgap of 1.17 eV [86]. The bandgap of the perovskite may also be adjusted through modifications to the lattice structure by varying the radius of the A-site ion. The MA^+^ replacement also impacts the bandgap with FA^+^ and Cs^+^ at A-site. FA^+^ can be introduced at the A-site to produce a NB perovskite, presumably due to FA^+^ having a slightly greater ionic radius than MA^+^ [87,88]. By adding various concentrations of FA^+^ to MAPb_0.75_Sn_0.25_I_3_, Jen and coworkers achieved enhanced stability without compromising the excellent NB PSCs efficiency on an Sn basis [67].

### 3.3. Optical Absorption and Bandgap Tuneability

Low-dimensional perovskites’ wider range of optoelectronic properties compared to their 3D counterparts is attributed to their structural diversity [89,90]. The optical bandgap of low-dimensional materials can be readily adjusted through dimensions and compositional engineering alterations. This characteristic proves advantageous in extending both emission and absorption wavelengths [43]. The ability to control the optical bandgap by changing the perovskite composition is essential for tandem applications. Compound engineering a just the bandgap from 1.2 to 2.3 eV [91,92]. The pure 2D perovskites, such as PEA_2_PbI_4_ (2.36 eV) and BA_2_PbI_4_, exhibited a significant bandgap (2.24 eV) [93,94]. Quasi-2D PEA_2_MA_n−1_PbI _3n+1_ with n = 40 has a bandgap of 1.52 eV, which is very close to 3D MAPbI_3_ (1.57 eV), as shown in Figure 7b [95].

**Figure 7 nanomaterials-13-01886-f007:**
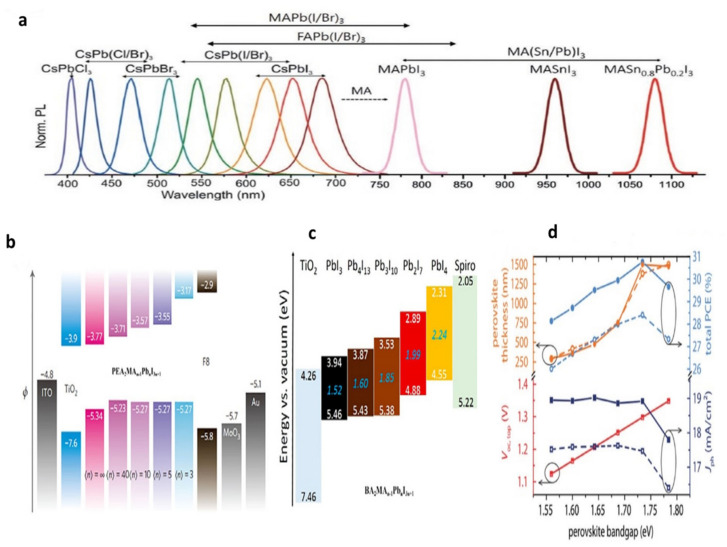
(**a**) Different metal halide perovskite materials’ normalized PL emissions [92]. Copyright 2018, John Wiley and Sons. (**b**) PEA_2_MA_n−1_PbnI_3n+1_ electronic band configuration that has various n-values [95]. Copyright 2021, John Wiley and Sons. (**c**) The band energy of the PEA_2_MA_n−1_PbnI_3n+1_ perovskite compounds is demonstrated schematically [95]. Copyright 2021, John Wiley and Sons. (**d**) The bandgap of monolithic perovskite/silicon tandem cells with an n–i–p (dashed lines) or p–i–n (solid lines) top cell is compared [96]. Copyright 2018, John Wiley and Sons.

This highlights how the structure of quasi-2D materials can mimic that of 3D perovskites when the number of layers is sufficiently high. The projected drop in bandgap with increasing n for BA_2_MA_n−1_PbnI_3n+1_ The findings in Figure 7c align with outcomes observed in PEA-derived perovskites, wherein the energy levels range from 2.24 eV (n~1) to 1.52 eV (n~ꝏ) [94,97]. The optical bandgap of 1D GAGeI_3_ (2.7 eV) is comparatively higher than that of 3D MAGeI_3_ (1.9 eV) in germanium-based perovskites. This is due to the weakened orbital overlap caused by the gradual spatial separation of inorganic frameworks due to the replacement of MA with GA [98]. The thickness of the perovskite absorber layer is another significant component in determining the maximum achievable photocurrent in the top cell. For a monolithic structure, the ideal perovskite layer thickness is depicted in Figure 7d, depending on the bandgap of an upper cell. The optimal perovskite layer thickness for a 1.73 eV bandgap is ~1 µm, which presents significant experimental difficulties while maintaining a high-quality material.

Consequently, in real-world applications, there is a trade-off between effective carrier collection, which needs good electrical quality throughout the film, and total light absorption at the initial light pass. Electro-optical studies must first be conducted to discover the constraints of a given perovskite deposition method [21]. Multiwavelength photoluminescence (PL) mapping can provide further insight into the material quality. When used in semiconductor materials, PL can reveal electronic flaws. Therefore, faults in particular cell layers can be linked to processing conditions or substrate surface state, provided the tandem layer stack is characterized as it moves through the process flow. Nonradiative recombination may be lowered by increasing the effectiveness of the perovskite layer as an absorber through a deeper comprehension of its growth process [99]. Combining photoluminescence and electroluminescence allows us to learn more about the tandem device’s limiting factors: PSCs photovoltaic and charge carrier transport efficiencies could be mapped with hyperspectral luminescence imaging. Combining photoluminescence and electroluminescence allows us to learn more about the tandem device’s limiting factors. Hyperspectral luminescence imaging may map PSC photovoltaic and charge carrier transport efficiencies [100]. As a result of oscillations in the interfacial resistance, low FF is found in device regions with an inefficient collection of photogenerated charge carriers. When considering contact materials based explicitly on the composition of the absorber material, the situation could be elevated by incorporating superior interface engineering and energetics.

## 4. Tandem Configuration

TSCs can attain superior efficiencies in comparison to single-junction devices. This is achieved by absorbing solar photons with higher energy in a top-cell material with WB. The photocurrent produced by the material in the top cell could reach a greater voltage than that of the solar cell below it; its absorbance coefficient is more significant, but its bandgap is less. PSCs are linked with c–Si, copper indium gallium selenide (CIGS) solar cells in tandem arrangement to raise the PCE of single-junction solar cells above the theoretical limit stated by SQ. It is possible to build a TSC in several different ways, with each method determined by how the junctions connecting the cells at the top and bottom are connected electrically. The efficiency of a one-junction solar cell is limited by the efficient use of only photons having energy close to the forbidden energy gap. Spectrum loss due to the restricted optical sensitivity of semiconductor absorbers accounts for most unusable solar energy. Photons with energies larger than E_g_ could be absorbed by a semiconductor, generating carriers with energies greater than the lattice one. The hot carriers in single junction solar cells induce hot carriers electromotive force with polarity opposite to the photovoltage resulting from electro-hole pair creation [101,102,103]. Solar cells’ efficiency is reduced due to light-induced carrier heating [104]. Coating silicon solar cells with semitransparent in infrared light region thin perovskite layers absorbing high energy photons can significantly reduce the detrimental effects of heated carriers [105].

Photons with energies below E_g_ cannot be used. Since the open-circuit voltage (V_oc_) and the short-circuit current (J_sc_) are the two factors that matter most when calculating a solar cell’s PCE, the absorber’s E_g_ is paramount. Overcoming the Shockley–Queisser limit has been suggested and developed using various techniques, such as multiple exciton synthesis, hot carrier collection, intermediate band construction, and tandem design [8,106,107,108,109]. The application of metallic nanoparticles leads to the increased efficiency of SCs due to the plasmonic effect [10,110]. This became possible by using light trapping through the resonant scattering and concentration of light in arrays of metal nanoparticles or by coupling light into surfaces plasmon polaritons and photonic modes that propagate in the plane of the semiconductor layer. Metal nanoparticles can excite the localized surface plasmon resonance, leading to increased light absorption and increasing the efficiency of SCs [111]. A significant photocurrent increase induced by metallic nanoparticles was observed in perovskite SC with incorporated Au/SiO_2_ core–shell nanoparticles [112]. When silica-coated gold (Au@SiO_2_) nanorods are embedded in the interface between the hole transport layer PEDOT:PSS and the perovskite CH_3_NH_3_PbI_3_, the average PCE increased over 40% from 10.9% for PSCs without Au@SiO_2_ to 15.6% with Au@SiO_2_ [113]. According to Jack et al., this enhancement could be due to the reduction in the binding energy of excitons by plasmons, which eventually accelerates the dissociation of excitons at the interface with the electron transport layer [10].

The optical and electrical independence of the four main topologies of TSCs varies. In a four-terminal (4T) configuration, the top and bottom cells must function as fully integrated devices to effectively merge their respective power outputs. The four-terminal (4T) arrangement features top and bottom cells that function independently. Longer wavelength incident sunlight may go from one cell to another without being blocked in 4T tandem arrangements (illustrated in Figure 8a) or be reflected by a neighboring bottom cell (Figure 8d). The configuration of series-connected monolithic tandems is considered the most desirable and complex (Figure 8b) [109]. The practicality of this approach lies in its ability to facilitate a more streamlined electrical connection while obviating top and lower cells’ need to have front and back electrodes, respectively. Both of these features make it possible to simplify the electrical connection. To successfully fabricate efficient 2T devices, one must overcome several challenges, including the following: (i) optical management within the tandem; (ii) matching of the currents in the upper and bottom cells; and (iii) the creation of recombination interfaces with minimal losses among neighboring cells or creating tunnel junction between them. The introduction of series-parallel tandem (SPT) configurations, as depicted in Figure 8c, presents a viable approach to amalgamate the power outputs of two cells while maintaining a comparable performance to 4T designs regarding regular energy yield. This is achieved by independently combining top and bottom cell strings and then connecting the voltage-matched strings in parallel.

### 4.1. 4T Tandem Solar Cells

The 4T TSCs exhibit optical linkage between their top and bottom layers while remaining electrically unconnected. This unique feature enables the top cells to operate as filters. This allows both cells to autonomously contribute to the maximum output power, as depicted in Figure 8a,c. Maximizing the efficiency of the upper and bottom cells at their respective current and voltage matching points is crucial for improving the performance of 4T TSCs. In TSCs, the top subcell is equipped with two transparent electrodes, one at the front and one at the rear. To function correctly, the front electrode needs a high level of transparency throughout the light-absorbing area and a high conductivity level. Since the top subcell must absorb visible light with high energy photons, the bottom subcell absorbs near-infrared (NIR) light; the rear electrode needs a high degree of transparency in the NIR region. It is the bottom line to construct an appropriate semitransparent electrode to optimize the efficiency of tandem devices. The initial investigation of a 4T PVK/Si TSC was conducted by Loper et al. in 2014. The front cell was fabricated utilizing MAPbI_3_, while the rear cell was constructed using a c–Si heterojunction [77]. The front cell produced a PCE of 6.2%, while the rear cell produced 7.2% using the transparent MoO_x_/ITO electrode in the 4T tandem cell. Ren et al. successfully achieved transmittance at long wavelengths utilizing a transparent MoO_3_/Au/MoO_3_ electrode. Oxygen annealing treatment was used to introduce a perovskite (PVK) thin film with reduced defect density, which allowed for the fabrication of a NIP-structured CH_3_NH_3_PbI_3_-based PVK top cell [114].

Transparent electrodes are commonly composed of TCOs, such as indium tin oxide (ITO), aluminum-doped zinc oxide (AZO), and indium zinc oxide (IZO), which are deposited through the process of sputtering. Different types of top cell architectures, such as NIP and PIN, for example, in an illustration of a typical 4T perovskite/Si TSC that has a homojunction Si bottom cell. A low-quality multi-crystalline Si bottom cell and a silver nanowire transparent electrode on a NIP-structured PVK top cell based on CH_3_NH_3_PbI_3_ were used to construct 4T perovskite/Si TSCs, which achieved an efficiency of 17%, as described by Bailie et al. [115]. The research team of Sargent showed that tandem devices with a configuration of 4T have demonstrated an efficiency exceeding 28.0% [116]. The utilization of opaque rear-mirror contacts in perovskite cells has been observed and exhibits an external quantum efficiency (EQE) typically greater than 80% close to the band edge. This indicates their effectiveness as single-junction cells.

In contrast, the EQE of semitransparent cells is typically closer to 70% while operating in the same spectral range. The perovskite film’s thickness was raised as a substitute, although the resulting films typically showed homogeneous morphologies and short carrier diffusion lengths. The aforementioned concerns were addressed through enhancements made to the solvent extraction methodology and the incorporation of a Lewis base. The outcome of this process resulted in the development of a perovskite film possessing a significant thickness, consistent morphology, and exceptional ability to remove carriers. The researchers have communicated the achievement of creating a partially transparent perovskite top cell that demonstrated a consistent PCE of 19.8%. The cell also demonstrated an average NIR transmittance of 85% in the range of 800–1100 nm. Furthermore, researchers have devised 4T MHP/c–Si tandem apparatuses, which demonstrated an overall PCE of 28.3% [117].

#### 4.1.1. Mechanically Stacked 4T Tandem Solar Cells

The two subcells that make up this structure are arranged in a manner analogous to that of the 2T tandem class. Conversely, although they are connected optically, they are not electrically dependent on one another, as shown in Figure 8a. Each subcells independent operation and optimization is made possible by having a distinct pair of terminals for each subcell. The mitigation of constraints on the selection of bandgap for the top cell is observed, resulting in reduced sensitivity of the device to fluctuations in spectral properties. As a result, it has been observed that 4T tandem cells can attain elevated levels of efficiency across a wide spectrum of bandgap values, ranging from 1.6 to 2 eV for the upper subcell. When paired with a c–Si lower subcell, an exceptional value of 1.81 eV can be achieved [96]. The presence of a matching layer, interface layer, or recombination layer among two subcells is deemed excessive. However, implementing this design necessitates using multiple transparent electrodes, potentially leading to increased parasitic absorption [118].

#### 4.1.2. Optical Splitting

This mirror is designed to selectively direct photons possessing high energies towards the perovskite subcell while simultaneously directing their low energies towards its silicon subcell [119]. A mono-crystalline (mc) SHJ cell is linked with a PSC made of MAPbI_3_, using an optical splitter to make the connection. As seen in Figure 9a, the perovskite cell is angled at a 90° angle towards the Si cell and oriented at 45° towards the optical splitter exposed to incident light. This configuration causes the reflected light to have a short wavelength and to be present perpendicularly on the lower bandgap cell [109,119,120].

The optical splitter possesses a cut-off wavelength in the optical spectrum (550 nm, 600 nm, and 640 nm). The Si cell can detect and absorb longer wavelengths of reflected light, whereas the perovskite cell is sensitive to shorter wavelengths. To attain an ideal cut-off of 600 nm, Zhao et al. employed an optical splitter composed of multi-layered dielectric oxides deposited through sputtering and possessed low (n~1.5) and high (n~1.9–2.2) refractive indices. Implementing a multilayer configuration reduces the extent of reflection losses at the interface between glass and air within 4T devices [121]. One of the benefits of utilizing a design of this type is that there is no requirement for additional transparent electrodes.

On the other hand, the higher cost of the optical splitter makes it more difficult for this tandem construction to be economically viable [96]. To mitigate the losses incurred due to free carrier absorption in the top and back contacts of perovskite cells, researchers recently employed an optical splitter in an experimental implementation of a bifacial design on 4T tandem configurations. The outcome of this experiment yielded a significant augmentation in the short-circuit current density (J_sc_) of the lower subcell of silicon, increasing from 15.15 to 33.5 mA/cm^2^. Furthermore, an increase in the production of electron-hole pairs was observed in the subcell, as mentioned earlier [122].

#### 4.1.3. Large Area Tandem Modules with Four Terminals

The slow progress toward semitransparent perovskite top cells with a high surface area may be responsible because most tandem devices still have an area of less than 1 cm^2^. Moreover, the enhanced sheet resistance of the transparent electrodes will result in heightened electrical losses for the tandem device, particularly as the surface area of the semitransparent perovskite cell expands. A novel architecture has been developed to fabricate high-performance 4T solar cells with significant area coverage. A thin-film PSC on top of a silicon solar cell is part of the current setup’s interconnected solar module [123,124,125]. The utilization of the module-on-cell structure in the tandem device comprising 4T perovskite–Si solar module resulted in an overall efficiency of 20.2% despite its small 4 cm^2^ aperture. The front and back electrodes’ sheet resistances were optimized, which helped achieve this result. To enhance the PV efficiency of a four-terminal solar module, it is imperative to reduce optical and electrical losses to a greater extent. As mentioned earlier, the aim may be achieved by lowering parasitic absorption by utilizing a highly transparent electrode and HTL in the NIR region. In addition, an inactive region in the upper perovskite module can be reduced between subcells by using efficient pulsed-laser ablation techniques, thereby enhancing the patterning process [126]. More research included a textured perovskite layer into an upper solar module to reduce optical losses further. It employed a refractive index matching layer within the air gap among the perovskite–Si stacks. The module-on-cell configuration only took up 4 cm^2^ of space and showed a 23.9% efficiency boost. A textured perovskite layer integrated into the top solar module facilitated this success [127]. Wide and narrow bandgap perovskite photoactive layers on flexible substrates may be developed by large-area blade-coating, as has been reported. Using an active material surface area of just 50 cm^2^ yields an efficiency of 15.3 %, and four-terminal tandem solar modules are constructed with optimized sub-junctions. The “module-on-module strategy” Combining a semitransparent perovskite solar module with a CIGS solar module can provide 4T tandem solar panels [124]. It is possible that flexible tandem solar modules consisting of all-thin-film layers could be viable, given that perovskite and CIGS absorbers can be integrated with flexible substrates made of either polymer or metal.

#### 4.1.4. Four-Terminal CIGS-Based Solar Cells

When 4T cells are taken into account, as is to be expected, the investigations provide an entirely different situation, with the engineering of the transparent conducting electrode (TCE) emerging as a key component of the research. There are numerous publications on 4T perovskite/CIGS solar cells, which contrast the monolithic configuration. In this context, it is important to highlight the work carried out by Shen and his colleagues. The researchers produced 4T cells with exceptional performance perovskite/CIGS by strategically integrating appropriately developed transparent electrodes and a multi-cation Cs_0.1_Rb_0.05_FA_0.75_MA_0.15_ PbI_1.8_Br_1.2_ perovskite absorber. The researchers utilized properly designed transparent electrodes for this purpose [128]. A 70 nm-thick coating of dense TiO_2_ and a 60 nm-thick film of mesoporous TiO_2_ were deposited onto an ITO bottom electrode of 100 nm thickness, and a perovskite top cell was produced. The top electrode, in contrast, was made up of a 180 nm-thick MgF_2_ AR coating on top of a 40 nm-thick IZO conductive layer and a 10 nm-thick MoO_x_ anode buffer layer. As a result, this phenomenon yields an excellent total NIR transmittance, exceeding 70%, accompanied by significant bandgaps of 1.62 eV and 1.75 eV in perovskites, resulting in PCEs of 18.1% and 16.5%, respectively. The research reports the attainment of 23.4% efficiency in a 4T tandem configuration, which exhibited remarkable stability against degradation caused by oxygen. This was accomplished by stacking a CIGS bottom cell with an efficiency of 16.5% and a bandgap of 1.13 eV mechanically on top of partially transparent PSCs with a high bandgap (1.75 eV). Bailie and his team utilized AgNWs as a back contact material to achieve a transparent perovskite-based subcell [115]. The overall efficiency was increased to 18.6% when paired with a CIGS rear subcell, which was 17% of the total. AgNWs were utilized by Lee et al. for the same objective but in a MAPbI_3_/CIGS device that had undergone complete solution processing [129]. Although this method is inexpensive, it only produces efficiencies of 10%. Device performance when employing AgNWs in perovskite-based devices is very unpredictable because the process is complex and involves solvents.

The study conducted by Gharibzadeh and colleagues involved the combination of a bulk 3D double-cation FA_0.83_Cs_0.17_Pb(I_1−y_Br_y_)_3_ structures with a 2D passivation agent based on n-butylammonium bromide (BABr) to create a 2D/3D perovskite heterostructure. This approach aimed to enhance the performance of 4T perovskite/CIGS TSCs. The result was a much higher V_oc_ and PCE values [130]. Changing the bromide concentration allows the 3D perovskite absorber layer’s bandgap to be controlled, ranging from 1.65 to 1.85 eV. The study determined that the stand-alone perovskite top cell exhibited a maximum stabilized PCE of 17.5% at a bandgap value of 1.65 eV.

Conversely, PCE values were significantly lower for bandgap values exceeding 1.74 eV. The researchers observed 4T TSCs by integrating perovskite cells mentioned earlier with CIGS bottom cells (E_g_ = 1.13 eV, PCE = 21.2%). The results demonstrated an overall tandem PCE of up to 25.0%.

#### 4.1.5. 4T Perovskite/Si Tandem Solar Cells

The PV industry typically uses crystalline silicon (c–Si) as its primary material. Perovskite-silicon TSCs have been more efficient than single-junction c–Si solar cells since 2018. The industry has been pairing perovskite-based TSCs with c–Si, as the silicon solar cells might benefit significantly from bottom-cell technology. This approach represents a viable strategy for commercializing PSCs in the current PV industry [131]. The subcells’ maximum power output is achieved in filtered and unfiltered 4T TSCs because of the lack of electrical connection between the subcells. Each subcell can independently increase its power conversion efficiency since current and voltage matching are unnecessary. Theoretically, 4T TSCs can achieve a PCE of 46% [108,132]. Fewer constraints on texture, cell polarity, processing technique, and temperature must be satisfied by other cells before a subcell may be created. This is performed to enable individual treatment of each subcell as a distinct entity. There is a significant reduction in the requirements of top and bottom cells in 4T TSCs. The remark implies that a simple method to enhance the PCEs of TSCs is to use high-performance double-sided texturing Si cells or PSCs produced at high temperatures. As a result, while other Si solar cells might be used as bottom cells, SHJ solar cells with double-side roughness and the greatest PCE would be the best option for 4T TSCs. The research team headed by Ballif reported the initial endeavor to attain MHP/c–Si TSCs [77]. To demonstrate this, they constructed a 4T tandem device with a MAPbI_3_ top subcell and a c–Si heterojunction bottom subcell, achieving a PCE of 13.4% as far as top and bottom subcells are concerned; 6.2% and 7.2% efficiencies were reported for top and bottom, respectively. Snaith et al. has presented their findings on developing high-performance 4T tandem cells incorporating a c–Si heterojunction cell. The resulting PCE of the tandem cell was measured and was found to be 22.4%. The authors have also demonstrated the successful fabrication of a mixed cation (FA/Cs) metal halide perovskite (MHP) solar cell, which exhibits high crystallinity and photostability in terms of composition. The optical bandgap of this MHP solar cell was determined to be 1.74 eV [133]. Recently, Ašmontas and his team explored the PV characteristics of a triple cation perovskite/silicon tandem SC with a four-terminal Cs_0.06_(MA_0.17_FA_0_._83_)_0.94_Pb(I_0.83_Br_0.17_)_3_ layer-based perovskite cell integrated on an industrial n-type monocrystalline bifacial PERT silicon SC [134]. The perovskite layer is very effective at absorbing visible light and is just slightly transparent in the infrared, as seen by its transmittance spectrum. In the 800–1100 nm wavelength region, the transmittance is considerably larger than 80%. The best PSCs have an open circuit voltage of 1.11 V, a short current density of 23.6 mA, a fill factor (FF) of 74%, and a PCE of 19.4%. The data for the bottom cell are as follows: 71% FF, 7.2% PCE, 15.8 mA short-current density, and 0.64 V open-circuit voltage. Thus, the 4T perovskite/silicon TSCs’ overall PCE of 26.6 % is substantially more significant than the efficiency of each subcell.

#### 4.1.6. Four-Terminal Perovskite–Perovskite Tandems

All-perovskite tandems’ design and processing flexibility are revolutionary compared to existing tandem technologies. The bandgaps of the upper and lower cells in any perovskite tandem may be modified to meet the most demanding design requirements and obtain the best possible performance. Additionally, the fabrication of the subcells may be accomplished with a low thermal budget, which helps to reduce the overall manufacturing cost [18,72,116,135]. Consequently, using all-perovskite TSCs might be a potentially fruitful technique to generate high-efficiency tandems that are low-cost, simple to manufacture, and lightweight. In 2015, Li et al. [136] constructed a 4T TSC using perovskite materials by stacking MAPbBr_3_ on top of MAPbI_3_. The HTL serves as the top cell’s means of facilitating charge transfer, and they utilize a composite material consisting of carbon nanotubes and PMMA. They opted for an ultrathin grid of gold metal as the transparent electrode. Transparent electrodes and high parasitic absorption in the HTL limited the top cell’s NIR transmission to 45% of its theoretical maximum. As a direct consequence, the performance of the TSC was reduced to 9.46% in PCE. The bandgaps of the upper and lower cells were not what would have been ideal for a highly functional TSC. Later, Jen and colleagues announced a 4T 19.08% PCE all-perovskite TSC that combined MA_0.5_ FA_0.5_ Pb_0.75_Sn_0.25_I_3_ with PSCs with a small bandgap (1.33 eV). Zhao et al. constructed a perovskite bottom cell with a NB to improve the NIR response [85]. Perovskite absorber layer thickness and grain size increased by adjusting the concentration of the precursor, resulting in better electronic properties. The Sn–Pb PSC’s NIR spectral response with a NB was better, resulting in a validated PCE of 17.01%. The increased light absorption was caused by the traits that these cells exhibited. Integrating a PSC with a WB showed a complete perovskite TSC with a 21% PCE. In 2019, Tong et al. [137]. The study established additional research involving the utilization of guanidinium thiocyanate (GuaSCN) in a perovskite precursor solution with a NB (FASnI_3_)_0.6_(MAPbI_3_)_0.4_ to enhance the structural and electrical characteristics of solar cells. As an optical filter, bandgap 1.63 eV semitransparent Cs_0.05_FA_0.8_MA_0.15_PbI_2.55_Br_0.45_ MHP solar cells were employed to achieve a PCE of 25.4% and a sustained efficiency of 25.0%.

### 4.2. Two-Terminal (2T) Perovskite–Silicon Tandem Solar Cells

Matching currents across subcells and ensuring processing compatibility across each layer and interface is crucial; obtaining monolithic 2T TSCs is technically more difficult than obtaining 4T TSCs. A 2T configuration characterizes the TSCs. It is hypothesized that 2T tandem cells will be more efficient than their 4T counterparts due to the reduced optical parasitic absorption and scattering at subcell interfaces. For this reason, 2T all-perovskite TSCs should be prioritized. To construct a 2T tandem device, a TCE with a metal grid and a transparent recombination layer is required.

On the other hand, a 4T tandem device necessitates using three TCEs, along with a physical gap separating the top and bottom cells. Employing low-temperature sputtering and atomic layer deposition, a TCO that includes ITO, hydrogenated indium oxide, AZO, and indium-doped zinc oxide (i-ZnO) could be created for the top TCE [138,139,140,141,142]. As a recombination layer, the top TCE material used is contingent on the feasibility of processability since the recombination layer needs qualities comparable to those of the top TCE. The previously discussed requirements must be satisfied for the device to function correctly. A layer of interconnectivity links the subcells electrically in series. The intermediate layer connecting the subcells is a tunnel recombination junction with low resistance, WB, high doping, or a transparent conductive oxide layer. The 2T architecture has various benefits over its counterpart. Transparent electrodes are needed in a 2T architecture, which minimizes the deposition processes and speeds up and lowers the cost of processing. In addition, having fewer electrode layers aids in reducing parasitic losses. In order to develop a 2T TSC that is affordable, commercially scalable, and incredibly efficient, significant research has been carried out to commercialize this technology within the next ten years [72,143,144].

Nevertheless, the 2T structure introduces a plethora of challenges in terms of processing. Kirchhoff’s law states that the aggregate device voltage equals the accumulated subcell voltages. Furthermore, the subcell that generates the least current limits the total current going through the TSC. This is because the subcells are linked together sequentially. To get the most out of a 2T TSC, it is essential to balance the current flowing through its numerous subcells. As a direct consequence, the bandgap and the absorber layer thickness in each subcell need to be meticulously optimized [145].

McGehee et al. presented a study wherein a monolithic 2-T perovskite/Si tandem device was developed by utilizing an n++/p++ Si tunnel junction on an n-type Si wafer in combination with an n–i–p perovskite subcell [146]. To achieve this, an n-type Si wafer was used. The n–i–p perovskite subcell of early PSCs used a MAPbI_3_ absorber and an electron-transport layer of mesoporous TiO_2_ that had been processed at high temperatures. TSC performance is inadequate because the Si subcell has neither been surface-textured nor has its p-type front been surface-passivated. As a result, the TSCs could only achieve a PCE of 13.7% at a voltage of 1.65 V. In a groundbreaking work by McGehee and colleagues, a low-temperature produced NiOx hole-transport layer was used with a p–i–n perovskite subcell. ITO was used as the recombination layer and placed on top of a flat-topped SHJ subcell [147].

An ALD-coated oxide layer was employed in this research to prevent the perovskite layer from being damaged during the ITO top electrode deposition. Specifically, the SnO_2_/ZTO bilayer technique was utilized, previously used within the interconnecting layers for all-perovskite tandem devices. The enhanced stability of this particular perovskite enables it to endure the application of a buffer layer of tin oxide via atomic layer deposition is being considered. Sputtering could be used to deposit a transparent capping electrode on top of the layer, which also inhibits shunts and has minimal parasitic absorption. The endurance of perovskite devices under a damp heat test at 85 °C and 85% relative humidity for 1000 h is attributed to a diffusion barrier in the window layer. This contributes to increased perovskite device’s thermal and environmental durability. For a 2T tandem device with a 1 cm^2^ active surface, it was proved that an efficiency of 23.6% could be achieved while maintaining a J_sc_ of 18.1 mA cm^–2^. Fluorinated ammonium’s electropositivity at its -NH3+ terminals can be significantly improved by increasing the distance between F and -NH3+. The outcome of this phenomenon is robust adsorption onto the anti-site defects of I_A_ and I_Pb_ with negative charges. These calculations were based on the work of Liu’s group, which conducted theoretical research [148]. The efficiency of inverted PSCs with a 1.68 eV bandgap is an incredible 21.63%, setting a new record. This is a significant accomplishment. In addition, a flexible PSC and a 1 cm^2^ opaque device both provide the greatest PCEs, 21.02% and 19.31%, respectively. Another research group reported that the perovskite/Si 2T tandem methodology resulted in a remarkable PCE of 29.15% when evaluated at the 1 cm^2^ device scale. Additionally, the device had a FF of 0.778, a J_sc_ of 19.75 mA cm^–2^, and a V_oc_ of 1.897 V [149]. Remembering that the Jsc values of each subcell in a 2T TSCs need to be well-matched for the cell to perform at its highest possible efficiency.

#### 4.2.1. Two-Terminal Perovskite–CIGS Tandems

In addition to silicon solar cells, CIGS solar cells may be used as bottom cells in hybrid TSCs. These cells have an ideal band gap of ~1.08–1.15 eV, necessary for today’s most cutting-edge, high-efficiency devices. The bandgap of CIGS, which is a direct-bandgap semiconductor, may be continuously adjusted within a range of 1.00–1.67 eV by changing the Ga/(Ga + In) ratios. In contrast to silicon thin films, which need an absorber layer thickness of over 200 µm, the absorption coefficient of CIGS can reach 10^5^ cm^−1^, low-temperature coefficient of −0.32%/K, low material usage (~2 μm), which reduces the absorber layer thickness to as little as 1–2 µm [150,151,152]. The capacity to adjust the band gap of CIGS and perovskites can yield tandem PCEs that are notably superior to those of perovskite/Si TSCs. This is the case because both materials have a tunable band gap. With this in mind, combining perovskite and CIGS could lead to a low-cost, thin-film tandem approach with high PCE. CIGS cells are anticipated to cost USD 0.34 W^−1^ for a manufacturing capacity of 1000 MW y^−1^ when the module PCE is 15% [153]. The use of CIGS and PSCs enables the processing of solutions, which makes it easier to produce TSCs that are both effective and affordable. Because CIGS and perovskite are both composed of thin films, they can be manufactured on flexible substrates. This enables roll-to-roll production, which has a high throughput and a cheap cost. The thin-film perovskite/CIGS TSCs, as they are initially fabricated, possess flexibility and affordability, and they hold promise for use in several cutting-edge industries, including portable electronics and building-integrated photovoltaics (BIPV).

The earliest monolithic perovskite-CIGS tandem devices were constructed in 2015, and their bottom cells were made using a solution rather than vacuum, leading to a noticeably flatter cell shape [154]. The recombination layer was an ITO of 30 nm to establish a connection between the two subcells. Since the perovskite layer is prone to degradation, the deposition process occurred directly onto the CdS layer, obviating the requirement for the CIGS devices to initially use an intrinsic zinc oxide (ZnO) layer. Figure 10a–c show an SEM image of the device and the associated JV curves, demonstrating that the J_SC_ produced by the tandem device was low, falling somewhere about 56% of that produced by the top perovskite subcell as a standard and CIGS cell as a shadowing element. The tandem device’s J_SC_ was lower than anticipated, which limits the transmission of incoming light to a maximum of 50% due to optical loss brought on by Al contact in the higher partially transparent perovskite cell. The maximum efficiency of the tandem device was 10.9%, which was much lower than the combined efficiency of the subcells. A perovskite top cell with a 1.72 eV bandgap and a CIGS bottom cell with a 1.04 eV bandgap was used in the configuration. Its inadequate structure significantly restricts the CIGS device’s performance, which is caused by the absence of intrinsic ZnO, which results in high parasitic absorption losses from the ultrathin metal electrodes. High series resistance due to inadequate devices made of CIGS and perovskite were found to have made contact with one another, contributing to the exceptionally low FF. Han et al. found practical solutions to these issues. The researchers used an ITO transparent electrode with a thickness of 100 nm rather than ultrathin metal electrodes. As a result of this layer’s high transparency, the light transmission was sufficient, and optical losses in tandem devices were reduced [155]. In addition, ZnO nanoparticle-doped transparent ITO electrodes provided excellent protection against moisture intrusion, enhancing the perovskite layer’s lifetime. To maintain high efficiency, TCO layers, called i-ZnO and boron-doped zinc oxide (BZO), were kept in the CIGS device structure. Chemical mechanical polishing was used to provide a uniform surface for the perovskite top cell, and an ITO buffer and recombination layer were used to compensate for the considerable vertical distance of the BZO layer. To enhance hole transportation due to a discrepancy in the work function of PTAA (−5.1 eV) and BZO (−4.0 eV), a polished ITO recombination layer was adjusted, resulting in superior ohmic contact. The study found that the efficiency of monolithic perovskite/CIGS TSCs increased significantly, reaching 22.43%. The improvement is attributed to the coexistence of a CIGS bottom cell and a perovskite top cell, which led to a noticeably increased J_sc_ of 17.3 mA cm^−2^ and an FF of 73.1%. Perovskite was discovered to have a bandgap of 1.59 eV, while CIGS had a bandgap of 1.00 eV.

In 2016, another team of researchers unveiled a device configuration that facilitates the production of inverted semi-transparent planar PSCs. These cells have an impressive open-circuit voltage of 1.116 V and a significantly higher efficiency of 16.1% [157]. Perovskite devices with the substrate configuration display encouraging thermal and photo-stability, with a temperature coefficient of −0.18% °C^−1^. The average transmittance of the device is 80.4% between 800 and 1200 nm, which is quite high. In 2017, Guchhait and colleagues conducted this work to examine the conductivity and transmittance performance of Ag/ITO and MoO_x_/ITO electrodes [158]. Even though the conductivity was comparable, substantial ITO delamination was seen for MoO_x_/ITO following exposure. This was because MoO_x_ is extremely sensitive to the oxygen in the surrounding environment. Consequently, the transparent electrode was chosen as a bilayer Ag (1 nm) and ITO (250 nm) structure. A tandem device with an overall efficiency of 20.7% was achieved by merging it with a Cs_x_(MA_0.17_ FA_0.83_)_(100−x)_Pb(I_0.83_Br_0.17_)_3_ perovskite top-cell that had a PCE of 16%. Semi-transparent PSCs with an ideal efficiency of 18.1% at a bandgap of 1.62 eV were produced by Shen et al. in 2018. When mechanically stacked in a tandem arrangement with a 16.5% CIGS cell, the resultant efficiency of the tandem arrangement is determined to be 23.9% [128]. By implementing optical management techniques, a notable mean transmittance exceeding 80% was attained within the 800 nm to 1200 nm range for a potential energy of 1.62 eV and within the 700 nm to 1200 nm range for a potential energy of 1.75 eV. The researchers emphasized the significance of interfaces as a key source of the O_2_-induced degradation of PSC. This is because recombination occurs at a much higher rate when an MA cation is present, which they observed to be particularly important. Perovskite/CIGS tandems have significant potential to exhibit an efficiency of greater than 30% using high bandgap perovskite, as proven by optical simulations carried out by this group. In 2019, Zhao and fellow workers conducted a comprehensive theoretical investigation to enhance the efficiency of TSCs [159]. Perovskite/CIGS TSCs with two terminals require the perovskite and CIGS layers’ thicknesses to be tuned to fulfill the present matching requirements. The thickness of the FTO was reduced to avoid reflection in order to boost the performance of two-terminal tandem cells, and a CIGS doping level of 1 × 10^18^ cm^−3^ was applied. Furthermore, the results indicate that augmenting the grain size of perovskite films could improve their quality by reducing the number of trapped states at the grain boundaries. When using two terminals, CH_3_NH_3_PbI_2_Br/CIGS TSCs may achieve an efficiency of up to 31.13% at their best [159]. In the year 2020, Nakamura and colleagues conducted research in which they prepared a variety of CIGS solar cells with E_g_ values differing from 1.02 to 1.14 eV [156]. Subsequently, through practical means, a spectrum-splitting mechanism was utilized to exhibit the outcome of utilizing lower E_g_ cells as the bottom cell in two-junction solar cells. In this study, the performance of a tandem cell design comprised of a top mixed-halide perovskite cell with a 1.59 eV energy gap and stand-alone efficiency of 21.0% and a bottom CIGS cell with a 1.02 eV energy gap and a stand-alone efficiency of 21.5% is reported. The tandem cell was fabricated using a 775 nm spectral splitting mirror with an aperture area of 1 cm^2^. The results indicate that the tandem cell exhibited an efficiency of 28.0%, as illustrated in Figure 10b. Kumar et al. conducted the computational modeling of two terminal Perovskite/CIGS TSCs in 2021 [160]. Based on simulations, this work introduces a unique perovskite, a CIGS tandem solar cell that offers long-term savings at lower costs and higher output. A simulated perovskite top cell is reported to possess a bandgap of 1.5 eV. A simulation on a CIGS bottom subcell with a 1.1 eV bandgap produced conversion efficiencies of 16.69% and 15.98%. The tandem configuration of the devices was assessed after the proper calibration of the upper and lower subcells. In order to determine the present matching point, modifications in absorber layer thickness were made at both the top and bottom. The ideal thicknesses for top and bottom subcells in tandem cells are 151 and 1000 nm, respectively. The tandem structure of CIGS/CdS/ZnO/Spiro/Perovskite/C-TiO_2_ proposed in this study has an open-circuit voltage of 1.646 V and a PCE of 23.17 %. A monolithic perovskite/CIGS TSC with a verified PCE of 24.2% was developed by a different research team in 2022 [161]. To identify the optimal device stack structure, optical simulations were used. The findings indicate that a significant amount of optical potential exists. The optimized structure attained a PCE of 32% and a short-circuit current density of 19.9 mA cm^−2^ while merely utilizing approximations of the authentic characteristics of the materials. When compared to CIGS and perovskite single-junction devices, which lose 9.7% and 5.6% of their energy, respectively, owing to temperature increases during field operation, the simulations showed that roughly 7% of the energy is lost in the tandem due to the temperature increase.

#### 4.2.2. Two-Terminal Perovskite–Perovskite Tandem Solar Cells

Two-Terminal perovskite–perovskite TSCs are solar cells that employ two perovskite-structured materials in a tandem arrangement to boost their overall efficiency. Additionally, the perovskite’s bandgap must be regulated to permit the coupling of two perovskite absorbers to build all-perovskite TSCs. Typical monolithic perovskite-perovskite tandem cell topologies are shown in Figure 11 Although no polarity has been shown to exist, p–i–n cells are more common.

All-perovskite TSCs have been developed to take advantage of both the tunability of WB subcells and the exceptional advance in single-junction device efficiency [162]. WB and NB perovskite subcells are used in all-perovskite TSCs and have become cutting-edge high-efficiency tandem devices, both theoretically and in experiments. All perovskite TSCs may undergo full solution and roll-to-roll processing, enabling low-cost manufacturing development, as shown with perovskite/CIGS TSCs. All-perovskite TSCs are thought to provide enormous potential for commercial breakthroughs due to recent advancements in single PSCs that are cheap, highly efficient, and large-scale [163,164,165,166,167]. Stacking a PCBM/MAPbI_3_/PEDOT:PSS/ITO bottom subcell alongside an FTO/TiO_2_/ MAPbBr_3_/PTAA(P3HT) top subcell, Heo et al. produced the first 2T perovskite/perovskite thin-film solar cell [166]. The Li-TFSI and t-BP additives in the HTM layer created a high-conductive recombination layer. The low current density caused by combining two WB absorbing layers capped the PCE at 10.4%. A high internal electrical field must be maintained for the directly laminated tandem device to prevent solvent from penetrating the top cell during perovskite deposition. Developing perovskite/perovskite TSCs presents a significant challenge: producing Pb–Sn perovskite materials with a high degree of NB and a recombination layer that exhibits excessive transmittance, excellent conductivity, and processing compatibility. Further, to use the near-infrared (NIR) spectrum in a tandem arrangement, the bandgap of the bottom cell should be smaller. MASn_1−x_ PbxI_3_ (0 < x < 1) was discovered to have strong NIR (700–1000 nm) photoluminescence (PL) emission characteristics by Stoumpos et al. [81]. To obtain a minimum bandgap of 1.2 eV, the Sn concentration must be between 60–80 mol%, based on the total amount of Pb and Sn [88,168]. To advance the efficiency of all-perovskite TSCs, a reduced bandgap perovskite material with a larger absorption spectrum was direly necessary. The absorber bandgap narrows once further when increased quantities of Sn are present. The observed phenomenon enables the diverse compositions of Pb–Sn mixed perovskites to exhibit moderate band gaps within the 1.22–1.25 eV range [82]. Simply the mixed cation halide perovskites of Sn–Pb composition have demonstrated reduced bandgaps appropriate for utilization in TSCs [169]. The exhibition of perovskite cells with NB and high efficiency, frequently called Sn–Pb perovskite cells, is a frequent additional problem linked with developing two-terminal perovskite–perovskite tandem. Eperon and colleagues fabricated a perovskite-perovskite tandem with two terminals, utilizing an Sn-containing perovskite with a NB (1.2 eV) as the material for the bottom cell. This allowed for more efficient energy conversion [170]. The instability of the Sn-containing perovskite is the primary challenge connected with it. The underlying reason for the instability observed can be attributed to the inherent tendency of Sn^2+^ to undergo oxidation, resulting in the development of Sn^4+^. The use of SnF_2_ or metallic Sn particles in the perovskite precursor has allowed this problem to be solved over the course of several years [81,171,172].

In 2016, a Pb–Sn mixed perovskite (FA_0.75_Cs_0.25_Sn_0.5_Pb_0.5_I_3_) with a low bandgap of 1.2 eV was used to extend the spectrum absorption range of all-perovskite tandems to wavelengths over 1000 nm [170]. The efficiency of the 2T tandem device was 17.0%, and the efficiency of the 4T tandem device was 20.3% when the NB (1.2 eV) bottom subcell with FA_0.75_Cs_0.25_Pb_0.5_Sn_0.5_I_3_ perovskite was paired with WB top cells with 1.8 eV FA_0.83_Cs_0.17_Pb(I_0.5_Br_0.5_)_3_ or 1.6 eV FA_0.83_Cs_0.17_Pb(I_0.83_ Br_0.17_)_3_ perovskites. The interconnection between the bottom and top cells was established by employing layers of zinc–tin–oxide and SnO_2_. The uppermost cell was made up of a lead cation and halide composition. Sputtered ITO was applied to the connected layers to prevent any damage to the underlying perovskite subcell from this process. A highly efficient 2T tandem SC reaching 80% of the theoretical limit in voltage was developed by Rajagopal et al. [173]. Optical simulations were utilized to validate the current matching criterion and identify possibilities for further improving the current generation in tandem architecture. Pb–Sn binary perovskite (MAPb_0.5_ Sn_0.5_I_3_) with a low bandgap of 1.2 eV was used as the bottom subcell, and WB perovskite Cs_0.1_MA_0_._9_Pb(I_0_._6_Br_0.4_)_3_ with a bandgap of 1.8 eV served as a top cell. Current-voltage characteristics of the best-performing 2T TSC show an exceptional V_oc_ of 1.98 V, a J_sc_ of 12.7mA cm^−2^, and an FF of 0.73, resulting in a PCE of 18.4%. The reduction in defect density in low-band gap perovskites became the primary focus of subsequent research and development efforts. Recent studies have shown that by adding chlorine, CdI_2_, and guanidinium thiocyanate (GuaSCN), the defect densities in Pb–Sn mixed perovskites can be significantly reduced by adding metallic tin to the perovskite precursor solution or by post-depositing absorber films with MACl vapor, both of which impede Sn^2+^ oxidation [137,171,174,175,176]. Using Sn–Pb alloyed perovskite absorbers with a NB, Yan and his team published a series of studies on all-perovskite TSCs [85]. The bottom cells absorbed additional infrared light when FA_0.8_Cs_0.2_Pb(I_0.7_Br_0.3_)_3_ with a larger bandgap of 1.75 eV and MoO_x_/ITO electrodes were utilized in the top cell as opposed to FA_0.3_MA_0.7_PbI_3_ (1.58 eV) and MoOx/Au/MoO_x_ transparent electrodes. Due to these enhancements, the PV performance of the NB (1.25 eV) (FASnI_3_)_0.6_(MAPbI_3_)_0.4_ bottom cell was much improved, while the top cell’s excellent light transparency of 70% over 700 nm was preserved [177]. Another research group accomplished a significant step forward with their research. NB (1.25 eV) absorber (FASnI_3_)_0.6_(MAPbI_3_)_0.4_ perovskite was improved through the incorporation of guanidinium thiocyanate (GuaSCN) into its structure and optoelectronic properties [137]. Applying an additive of 7% GuaSCN to NB, perovskite films resulted in a tenfold decrease in defect density and an increase in carrier lifetimes of more than 1 µs. Better film morphology with fewer grain boundaries and pinholes decreased surface recombination velocity to 1.0 × 10^2^ cm s^−1^ and increased carrier diffusion length to 2.5 µm. A stabilized efficiency of 20.2% was shown by a (FASnI_3_)_0.6_ (MAPbI_3_)_0.4_ perovskite treated with 7% GuaSCN. In addition to merging with WB PSCs, they achieved a PCE of 23.1% for 2T devices. Bandgap enhancement without a corresponding increase in Br content has also been achieved by using the bulky cation dimethylammonium in the perovskite composition, which has been used extensively in optimizing the WB top cell. Because of this, photo-stable perovskite compositions with a bandgap of 1.7 eV and minimal voltage losses have been produced. This has made it possible to create 2T tandem cells on flexible substrates that are 21.3% efficient [178]. The comparison of 2T/4T perovskite tandem solar cells is provided in Table 1.

#### 4.2.3. The Layer of Recombination in a 2T Tandem

Regarding the design and manufacturing procedures, the recombination layers present among the top and bottom cells in two-terminal tandems are among the layers that provide some of the biggest obstacles. To link two distinct cell layouts, the layers must integrate electrons and holes with little voltage loss and transparency degradation. A recombination layer is required for the current to travel between the two sides of a 2T tandem. As mentioned earlier, the process is achieved through the facilitation of the recombination of opposing carriers within the subcells, which occurs in a heavily doped layer. The ideal recombination layer would minimize optical and electrical losses while boosting electron and hole recombination between two subcells [178,179,180]. Due to the need for a lattice match between the subcells and the tunnel junctions, the range of design possibilities for conventional III–V multi-junction solar cells is severely constrained. In high-efficiency III–V semiconductor TSCs research, a tunnel junction can be built as the recombination layer by heavily doping the interlayers between the subcells with n- and p-type elements. It will become less critical to have the valence band maximum and conduction band minimum of the p-type and n-type semiconductors match up in energy. It will also minimize the undesirable nonradiative recombination between the subcells [181,182]. Perovskite tandems are exempt from this constraint, which paves the way for innovative designs of recombination layers. Shunt paths employing the top cell can be drastically reduced by creating a thick recombination layer with poor lateral conductivity. Previous decades have seen several publications demonstrating the usefulness of thick conformal recombination layers in reducing shunts in 2T tandems [178,179,180].

An effective recombination layer must have a high carrier recombination rate, excellent optical transparency, good ohmic contact, and seamless interaction with other component manufacturing processes. To date, one of the most common options for connecting the subcells of 2T all-perovskite TSCs is to use a single TCO layer with a significant doping concentration (~10^20^–10^21^ cm^−3^). For some applications, an additional layer of ultrathin metal (~1 nm) was developed to improve the efficiency of charge recombination [171,174]. TCOs, including indium tin oxide (ITO), hydrogenated indium oxide (IO: H), AZO, and IZO, are commonly used as conductors in perovskite tandem solar cells [77,125,133,179,183,184,185]. Due to their outstanding optoelectronic capabilities, TCOs, such as ITO and IZO, appeal for a recombination layer in organic TSCs. The 2T perovskite/SHJ tandem device of Albrecht et al., including a SnO_2_/ITO recombination layer, shows a high V_oc_ of 1.78 V and a stable efficiency of 18.1% [186]. Werner et al. utilized a recombination layer of indium zinc oxide (IZO) and an ETL of PCBM to fabricate tandem devices featuring a 2T perovskite/SHJ architecture. The stability of the SHJ bottom cell has been guaranteed by fabricating the semitransparent perovskite top cell at low temperatures. The individual showcased the method of generating semitransparent PSCs through low temperature means, resulting in up to 14.5% efficiencies. Subsequently, the previously mentioned methodology was employed to synthesize integrated perovskite/silicon heterojunction TSCs, yielding efficiencies of 21.2% and 19.2% for cell dimensions of 0.17 cm^2^ and 1.22 cm^2^, respectively [187].

Several optical simulations conclude they cause optical losses of around 1 mA cm^−2^. Since the TCO and the bottom of the cell absorber contact have different refractive indices, the TCO suffers from issues including free carrier absorption at NIR wavelength and Fresnel reflection losses. Sahli and fellow workers used a perovskite–SHJ tandem configuration with a recombination layer built of highly doped nc–Si:H to limit the effect of Fresnel reflection losses. More than 1 mA cm^−2^ was added to the SHJ bottom cell’s output after the nc–Si:H recombination layer enhanced index matching with the c–Si absorber. We can be grateful to this improvement for the reduction in reflection loss compared to the ITO version. These losses have been reduced by showcasing a refined recombination junction that uses nanocrystalline silicon layers. This combination has been used to increase the photocurrent of the bottom cell in monolithic heterojunction tandem cells made of perovskite and silicon by more than 1 mA cm^−2^ when the front side is flat [188]. Nevertheless, the low-bandgap nc–Si:H recombination layer was shown to be responsible for the parasitic absorption that led to the optical losses. Mazzarella et al. successfully resolved the issue of low light coupling to the bottom cell by introducing oxygen alloying into the nc–Si:H layer. This resulted in creating a WB gap nc–SiOx:H recombination layer [189]. Mailoa et al. had successfully developed the initial 2T tandem featuring an all-silicon tunneling junction. Further enhancements are anticipated to mitigate voltage loss compared to the TCO–Si tunneling junction [146]. A 1 cm^2^ monolithic perovskite/silicon multijunction solar cell with two terminals and a V_OC_ of 1.65 V was successfully developed. Using perovskite as the current-limiting subcell reliably yielded a PCE of 13.7%. The device architecture faced significant obstacles that must be addressed to attain efficiencies exceeding 25%. Significant gains can be made by switching out the Spiro-OMeTAD layer for a hole transport material with a more considerable band gap. The quality of the perovskite absorber will improve as a result of this replacement. The desired outcomes were attained through utilizing specialized furnaces to produce the Si subcells and the enhancement of surface passivation methods applied to both the front and back surfaces of the Si subcells.

The recombination layers of 2T perovskite TSCs, predominantly comprising soft organic and hybrid materials, must possess high transmittance, high conductivity, and processing compatibility to ensure optimal performance. Consequently, the recombination layer suitable for this scenario ought to maintain the flexibility that can accommodate the soothing properties of the underlying cells. To bridge the gap between the two subcells, Jiang et al. recommended utilizing a Spiro-OMeTAD/PEDOT:PSS/PEI/PCBM:PEI multilayer organic recombination layer. The outcome yielded a V_oc_ value of 1.89 V and an overall efficiency of 7.0% for the 2T all perovskite TSCs [190]. Protecting the perovskite layer’s structure beneath the recombination layer is essential; all its component elements were produced by utilizing orthogonal solvent processing techniques at reduced temperatures. Subsequently, a brief annealing process was conducted. The recombination layer, possessing an average thickness of approximately 200 nm, can efficiently extract and convey carriers generated in the subcells while safeguarding the uppermost perovskite layer from any possible solvent infiltration. Using a solution-based method, a recombination layer composed of PEDOT:PSS and ITO NP sublayers was developed by McMeekin et al. [191]. Using a spin-coating technique, the Spiro-OMeTAD layer was first coated with the PEDOT:PSS layer. Subsequently, the indium tin oxide nanoparticles (ITO-NPs) were applied onto the PEDOT:PSS layer via spin-coating using a dispersion solution. The PEDOT:PSS layer exhibited recombination and partial solvent barrier characteristics. The ITO NPs facilitated enhancements in the recombination process, thereby augmenting carrier transport efficiency. The 2T perovskite/perovskite tandem devices were able to attain a PCE of 15.2% in a steady state attributed to the recombination layer. Advantages and disadvantages of 2T and 4T tandem solar cells are summarized in Table 2.

## 5. Perovskite Tandem Solar Cells: Challenges

When PSCs are combined with other cutting-edge PV technologies, all the benefits and drawbacks of both are amplified. Perovskite-based TSCs have a lot of potential in their PCE, which is a very encouraging development. The PCE of solar cells is among various factors that contribute to their suitability for widespread use, in addition to their huge processing area, excellent throughput, low environmental impact over time, and other relevant considerations. The factors that facilitate a smooth transition from laboratory research to industrial production are of significant interest, given the established validation of silicon and CIGS solar cells in the current market. The sustainability and scalability of PSCs remain a subject of inquiry. To be commercially viable, the perovskite top cell must satisfy the same field assurance standards as the silicon bottom cell for a silicon-based tandem solar panel. Furthermore, wafers made for industrial manufacturing must be used to make tandem cells, and only minimally expensive technologies and materials must be used. Scaling up devices to module size is just one of the many obstacles to commercializing this technology; increasing the operating lifetime to 25 years is another key hurdle.

### 5.1. Device Structure

The two subcells of 4T tandems are completely separated electrically. The device can still function overall, although with reduced power production if one of the subcells is not functioning. The devices within a 2T tandem are susceptible to destruction in the event of a failure in either subcell or the recombination layer, owing to their direct interconnection. Creating vital semitransparent perovskite front cells is a significant challenge for 4T tandem devices. Since there are more transparent electrodes and interfaces in a 4T tandem, it is more susceptible to inefficient reflection and parasitic absorption than a 2T arrangement. Because of this, two-terminal topologies predominate in the literature when discussing tandem devices. In addition, we believe that the 2T architectures will continue to be the norm.

When fabricating an efficient monolithic multi-junction solar cell, several specific problems must be overcome in addition to optimizing each subcell. The first need for effective 2T device design is to pick band gaps that correspond with the current generated by each subcell. A band gap of 1.73 eV is considered optimal for the top cell in both Si and CIGS. More than 40% efficiency may be attained theoretically with a top cell band gap of 1.65 to 1.85 eV. However, the performance reduction is insignificant when constrained by a tried-and-true bottom cell technology, such as Si or CIGS (E_g_ = 1.11 eV) [192,193,194]. Applying a 2T tandem connection to optimize high-quality Si is difficult, even when the optimum band gap is specified. Producing efficiencies more significant than the single junction efficiency of silicon or CIGS has been challenging because there have not been any readily available extraordinarily efficient high-band gap top-cells (beyond the III–V growth). Another significant challenge for efficient tandem design and optimization is ensuring process compatibility while layering fabrication. The top cell’s maximum processing temperature must tolerate the bottom cell and pn-junction of a substrate device, and the reverse is valid for a superstrate device. One can decouple the difficulties associated with temperature compatibility by linking two fully functional devices [192,195].

### 5.2. Long-Term Stability

After a quarter of a century of practical application, the pre-eminent silicon PV technology ensures an electricity output of more than 80% of its original PCE. This technology has shown exceptional endurance. CIGS and CdTe, two examples of thin-film PV technologies, have shown an impressive capability for long-term reliability [145,196]. PSCs have struggled with stability since their inception. Poor stability was the main reason Mitzi et al. abandoned their research on perovskite materials for PVs in the early 1990s [33]. However, a lot of work has been undertaken in the past five years. Numerous research groups have shown hundreds to thousands of hours of consistent data by adhering to standard degradation protocols [197,198]. To compete with these well-established PV technologies, perovskite-based TSCs must meet or exceed their standards for efficiency and durability. Perovskite-based TSCs with a 2T configuration exhibit restricted operational stability for a few months. This duration is when the solar cell can maintain its stability while being exposed to the sun’s illumination and generating its maximum output [179]. The instability of perovskite materials and single-junction perovskite devices have been the primary foci of previous studies.

In contrast, the 2T perovskite/c–Si tandem has not been studied to the same extent [199,200,201]. The enhanced management of anions in the perovskite compound improved stability. Following consistent illumination for 1000 h, a tandem device consisting of 2T perovskite and c–Si retained 80% of its initial efficiency. This was demonstrated in a study by Kim et al. [144]. A semitransparent PSC encapsulated in EVA glass–glass encapsulation the damp heat test (85 °C/85% relative humidity) performed by Bush et al. with negligible thermal and moisture deterioration [147]. According to results from experiments conducted by Aydin et al. on two-terminal monolithic PK/c–Si tandems in hot, sunny weather, the perovskite’s ideal bandgap may be less than 1.68 eV under typical test circumstances [202]. The 2T perovskite/c–Si tandem underwent outdoor testing at KAUST for one week. This report presents the inaugural outdoor performance analysis of the 2T perovskite/c–Si tandem. It is essential to point out that, as of yet, there has been no report of a 2T perovskite/c–Si tandem device passing the IEC61215:2016 standard (temperature 85 °C), low temperature (−40 °C) paired with high relative humidity (85%), and/or rapid change in temperature (200 °C h^−1^). As a result, instability may count as one of the main challenges to bringing perovskite/c–Si tandem to market. PSC instability problems can be attributed to internal variables, such as ion diffusion or migration, as well as environmental factors, such as sensitivity to oxygen and humidity. The impact of subcell failure on 2T and 4T TSCs varies depending on the mode of connection, with unique effects observed in each case. Due to the fact that the two subcells of a 2T TSC are connected in series, if either the subcells themselves or the recombination layer fail, all of the devices will be rendered useless.

#### 5.2.1. Material-Related Instability

There are many skeptics regarding the commercialization of this promising technology because of the perovskite material’s instability. Because of its soft-matter nature, the most popular MAPbI_3_ perovskite absorber has a bad reputation for tolerating heat, moisture, and light. These crystals are unstable, and the organic component in the crystal lattice may have something to do with this [203,204,205]. MAPbI_3_, a perovskite material, has a tetragonal phase at ambient temperature. Still, it undergoes a phase transition to a cubic phase at 60 °C, within the practical temperature range of a solar module. Under conditions of high temperature, the methylammonium cation was also observed to rapidly diffuse out of the cell [206]. Perovskite-based TSCs have serious difficulty at the material level because of the hybrid perovskite materials’ intrinsic resistance to water, heat, light, and oxygen.

In most cases, adding Cs can partially replace the volatile A-site organic cations (i.e., MA^+^, FA^+^) with stable WB perovskites [90,207]. It has been demonstrated that the structural stability of the perovskite material can be significantly improved by adding MA and/or FA. Using inorganic cations, such as cesium and, subsequently, rubidium, marked a significant step forward in the stability of materials. After hours of maximum power point tracking, cells with cesium and rubidium cations maintained 95% of their initial efficiency [23,165,207,208,209,210,211]. After a stability test, more than 85% of the initial PCEs were maintained by all of the CsPbI_3_ solar cells, indicating that they are reasonably stable. Zhao et al. found that the performance of a solar cell based on γ-CsPbI_3_ remained constant during a stability test performed without encapsulating material, even in the presence of extreme ambient conditions. One possible explanation is that the γ-CsPbI_3_ thin film is thermodynamically stable [212]. Potentially valuable for tandem structures, CsPbI_3_ solar cells’ total efficiency is still lower than that of hybrid perovskites with the same bandgap. More study is needed to improve the single junction device’s efficiency.

#### 5.2.2. Intrinsic Stability

Ion diffusion and migration are cited as reasons for perovskite devices’ intrinsic instability. Because perovskite materials are ionic conductors and have loose crystal structures, in addition to the presence of vacancies and other defects, ion migration is a phenomenon that is intrinsic to these materials. Light and/or heat conditions can speed up the loss of perovskite ions, which would cause the lattice to collapse and change the film’s composition and morphology once ion migration occurs in perovskite materials [213,214]. Perovskite crystals’ formation temperature is low, resulting in a relatively facile production process. Perovskite crystals exhibit a delicate and soft nature. Thus, they rapidly decompose when exposed to heat, even at moderate levels. Figure 12c illustrates the ion migration phenomena in organometal halide perovskites [215]. Mizusaki et al. discovered this behavior for the first time in 1983 [216]. According to the authors’ proposal, ionic conduction in halide perovskites was hypothesized to originate from the movement of halide-ion vacancies inside the perovskite lattice. Perovskite films are prone to undesirable band bending, interfacial interactions, and phase segregation due to ion migration during device operation.

The passivation of defects has the potential to inhibit the movement of ions. The hysteresis and instability of PSCs would result from ion movement inside perovskite materials when subjected to an electric field [217,218]. Since the hysteresis greatly impacts the reliability of the device’s photovoltaic performance under operational conditions, it is also seen as an obstacle to developing stable PSCs. The J–V hysteric properties of PSCs were first reported by Snaith et al. [219]. Hysteresis refers to the deviation in the forward and reversed scan current-voltage (J–V) curves: while scanning in reverse, the photocurrent drops exponentially before reaching a steady state, and while scanning forwards, the photocurrent increases exponentially before reaching a steady state. Hysteresis is significant since it is correlated with PSCs’ performance parameters and durability over time. Various studies have found that the hysteresis of multiple factors impacts PSCs, including scan rate, perovskite composition, particle size, and device structure [68,220,221,222,223,224]. Moreover, McGehee et al. found that the hysteresis in PSCs was very sensitive to J–V measurement parameters, such as pre-bias, scan rate, and scan direction [225]. PSCs’ photovoltaic performance was enhanced when exposed to a 1.2 V forward bias in constant light, but it was degraded when exposed to a 1.2 V reverse bias during light soaking. The photo-induced ion migration in perovskite caused by the applied electric field was thought to be the primary cause of the substantial impact of the pre-bias conditions on J–V measurements. According to the findings of this study, any ion migration in perovskites would not only impact the reliability of the J–V measurements but also damage the PSCs’ capacity to maintain their stability over the long run. Even though numerical modeling revealed that charge trapping and de-trapping were needed for hysteresis, the timeframe for charge trapping and de-trapping (picosecond) was too rapid to explain the slow hysteresis with a period of 1–10 s [220,226]. A possible cause of hysteresis is the movement of ions since the low frequency (1 Hz) enormous capacitance found in PSCs is thought to be strongly related to the J–V hysteresis and can be accounted for by the electrode polarization induced through ionic charge accumulation [227,228].

The normal thermal instability is generally brought on by the loss of organic components (MA^+^), resulting in many flaws at the grain boundaries, similar to humidity instability [229,230]. Attractive solutions to this issue include passivating grain boundary defects and optimizing the “A” cation with suitable materials (FA, Cs, Rb, and K). The thermal stability of inorganic perovskite has been demonstrated in several investigations to be superior to that of organic perovskite [23,231,232,233]. Stability is necessary for successfully commercializing 2T perovskite/c–Si tandem, since solar cells need to function in light conditions maintained over long periods. The majority of high-efficiency 2T perovskite/c–Si tandem top cells utilize mixed-halide hybrid perovskites as their active material [234].

### 5.3. Charge Transport Layers

Organic semiconductors were frequently utilized as charge transport layers in PSCs, which were problematic since they were quickly oxidized or absorbed water. The utilization of n-type fullerene PCBM has been observed to be an effective charge transport mechanism in PSCs. Conversely, it was determined that PCBM exhibited instability in the presence of ambient air, attributable to fluctuations in its chemical states or band structure. This discovery formed the basis for explaining why the device degraded over time [235]. The present layer has the potential to absorb moisture from the surroundings and corrode the transparent conductive electrode, specifically ITO, owing to the acidic nature of PEDOT:PSS. This, in turn, results in the deterioration of the devices. P-type of PEDOT:PSS was typically used as the HTL in the inverted structure [235,236]. Corrosive decomposition of MAPbI_3_ to PbI_2_ has been demonstrated in the presence of dopants and additives, such as 4-tert-butylpridine in Spiro-OMeTAD. However, annealing Spiro-OMeTAD causes it to crystallize, reducing contact with the perovskite layer [237,238]. The inclusion of lithium salt dopant within the Spiro-OMeTAD layer possesses the capacity to absorb water, thereby leading to device malfunction and potential water infiltration into the perovskite layer [239]. An opaque metal is frequently employed as the back electrode when working with single-junction perovskite cells. As per the conclusions drawn by several research teams, it has been suggested that the metal layer could potentially lead to degradation via one of three mechanisms: first, through corrosive reactions involving the byproducts of the perovskite absorber degradation; second, through redox reactions regarding the metal electrode and the perovskite absorber material; and third, through the migration of metal particles [240,241,242,243]. The low stability of PSCs can be traced back to the perovskite materials and interface, and many groups are still trying to improve the situation.

## 6. Perovskite Tandem Solar Cell Power Losses

Power loss is a crucial factor to consider when designing and making tandem devices. The ability to build highly efficient devices relies on having a solid understanding of the elements that affect the loss of power. Other than electrical losses, reflection, parasitic absorption, and another optical loss account for a considerable portion of total power losses. Parasitic absorption losses arise in tandem device structures when photons are absorbed in layers that do not generate photocurrent. Monolithic TSCs experience high electrical or resistive losses during operation, which causes current mismatch and lower efficiency.

### 6.1. Parasitic Absorption

The 2T and 4T c–Si/perovskite tandems benefited from the use of transparent electrodes for a number of reasons. T transparent electrodes used in the top PSC and bottom silicon subcells must have low sheet resistance and high transparency to allow for the transmission of a significant amount of light across a broad spectrum (UV-vis and in the NIR spectral range). The optical absorption at the TCO electrode, charge transport, and recombination layers generated parasitic absorption loss. Within these layers, photons that are absorbed do not make any contribution to the photocurrent. As a result, there is a loss in absorption due to parasitic factors. Therefore, a significant barrier to four-terminal tandem structures is the necessity for three transparent electrodes. The transparent electrodes’ free-carrier absorption is thought to cause a noticeable increase in parasitic absorption losses in the 850–1200 nm wavelength range. This effect is particularly pronounced for layers exceeding a thickness of 100 nm [244]. Numerous experiments using various materials to increase transmittance have been reported on transparent electrodes. Many research groups have demonstrated that electrodes based on AgNWs mesh can be produced by implementing spray coating and mechanical transfer deposition techniques [115,245,246]. An interaction between silver halide complexes and ions from the perovskite layer quickly raised concerns about the layer’s stability [179,240]. One of the well-established methods for minimizing parasitic loss is to reduce the layer thickness of transparent electrodes. This method is one of numerous that have been developed. The parasitic absorption and reflection were successfully reduced by reducing the thickness of the front ITO layers from 150 to 60 nm [247,248]. On the other hand, the optical simulation revealed that thicker ITO layers result in considerable parasitic absorption. The reduced thickness of layers decreases the FF owing to the elevated series resistance. Significant parasitic charge carriers cause both of these phenomena [147,249]. To minimize the effect of parasitic absorption, one other method would be to use electrodes that are not made of ITO and have a lower carrier concentration but a high carrier mobility IZO [154]. Using NiO_x_ HTL, phenyl-C61-butyricacid-methyl-ester (PCBM) ETL, and the MoO_x_/IZO front electrode, Sahli et al. were able to effectively manufacture a monolithic tandem with considerably decreased parasitic absorption [188]. There was a slight possibility that the charge transport layers underneath the perovskite layers could have been harmed when transparent electrodes were placed on top of those layers. This issue was fixed by adding buffer layers between transparent electrodes and organic charge transmission. In sputtering, MoO_x_ is commonly employed as a p-type contact and a buffer layer to safeguard the subjacent soft layers. MoO_x_ is a variant of the transition metal oxide (TMO) and was thermally evaporated [77,183,250]. As an alternative to MoO_x_, tungsten oxide (WO_x_), a more resilient metal oxide, was offered as a substitute. Atomic layer deposition might be used to deposit ZnO NPs and SnO_2_ thin films as an alternative option [147,157].

### 6.2. Reflection Losses

It is a verifiable truth that in both the 2T and 4T c–Si/per tandem configurations, the upper perovskite subcell must be traversed by light prior to its arrival at the lower silicon subcell. As a result, it is desirable to improve the intensity of NIR light that is losslessly transported from the top perovskite cell to the bottom silicon cell. TSCs experience a notable reduction in power output due to reflection loss caused by refractive index (n) mismatch among the various layers. This is especially true for light with a long wavelength. Furthermore, it is worth noting that the phenomenon of reflection losses can be attributed to alterations in the refractive index at the interface of the front surface (comprising the front electrode and free air) and the interfaces between the upper and lower subcell layers stack. A substantial fraction of the reflection loss can be ascribed to the refractive index mismatch among the air (n = 1), glass (n = 1.52), and the transparent electrode. Significant losses occur at the front electrode due to the deflection of a portion of the light away from the higher refractive index layers, resulting in no contribution to the photocurrent [251,252]. It is possible to minimize reflection losses by adjusting the refractive index of the TCO layers; as has been reported, the nc–SiO_x_:H layers in thin-film silicon TSCs were substituted with TCOs with n values ranging from 1.8 to 2 [253,254]. The results show reduced parasitic absorption and reflection losses in the red and infrared spectral ranges. The silicon bottom cell’s photogenerated current was also improved using the nc–SiO:H layer as an interference/recombination junction between perovskite/Si monolithic TSCs [255]. Another idea is to minimize the effect of the refractive-index mismatch within the tandem solar cells. For instance, by utilizing a multilayer stack in the tandem design that displayed spectrally selective transmission/reflection behavior, the J_SC_ was able to be enhanced by 0.82 mA cm^−2^, a significant amount [256]. Investigations have been conducted using polydimethylsiloxane (PDMS) films, which have been shown to have a refractive index (n~1.4) that is comparable to that of glass (n~1.52) and to be capable of significantly reducing reflection at shorter wavelengths. At 550 nm, the reflectance of the flat PDMS is 8.12%, which is considerably lower than the reflectance of bare c–Si, which is 38% using textured PDMS; this can be minimized even further [257].

It is highly recommended to use anti-reflective coatings to lower the reflection losses. Since both lithium fluoride (LiF) and magnesium fluoride (MgF_2_) are highly transparent and have a low refractive index, they are widely employed in anti-reflective coatings [165,186,208,258,259,260]. Incorporating a LiF layer on a monolithic c–Si/per thin-film solar cell by thermal evaporation allowed Albrecht and colleagues to decrease reflection losses at the air/ITO contact [186]. Adding a LiF layer as an anti-reflective coating resulted in a 1.5 mA/cm^2^ increase in the photo-generated current as measured in the bottom silicon subcell. Anti-reflective coatings also made it easier for the two subcells to show closed-matched currents. Under AM 1.5G spectra, the current values for the top perovskite and bottom silicon subcells were 14.7 and 14.0 mA/cm^2^, respectively. It is anticipated that similar to the case of amorphous silicon, plenty of research will be conducted in the near future regarding the specific advancement of wide-band antireflection coatings for thin-film technologies [261].

## 7. Conclusions

This review provided an in-depth assessment of the advancements made during this intense period, emphasizing the perovskite/silicon tandem solar cell advancement. Material characteristics, device designs, and basic working mechanisms have advanced perovskite development. Because of the limited number of layers, they can be produced cheaply, have great efficiency potential with low parasitic absorption, and be easily used in PV systems. Efficiency records have been broken rapidly due to the strengthening of tandem structures in combination with perovskite/contact materials. Lab-scale 2T perovskite/c–Si tandem efficiency is promising for commercializing perovskite/Si tandem technology.

Regarding prospective and actual application in industry, the monolithic two-terminal tandem cell is the way to go. The efficiency of commercially available Si bottom cells is quite close to their theoretical maximum. Multiple challenges, including performance improvement, the intrinsic instability of perovskite materials, the expense of acquisition, and scaling-up manufacturing, continue to obstruct the development of tandem devices. It has been shown that single-junction solar cells have intrinsic flaws; hence the benefits of multifunctional cells have been emphasized. The discourse has encompassed various categories of TSCs and their corresponding mechanisms: the two-terminal monolithic, four-terminal stacked, and optically splitting solar spectrums. However, this conversation will continue to be grounded in state-of-the-art research and the significant advances of recent years. We think it is realistic to aim for a demanding goal with a two- and four-terminal tandem configuration. Perovskite photovoltaics have a dedicated fanbase, and as a result, they are expected to make rapid advancements in the near future. The commercialization of solar energy as the most commercially viable energy source may be facilitated by using perovskite absorbers in TSCs.

## Figures and Tables

**Figure 1 nanomaterials-13-01886-f001:**
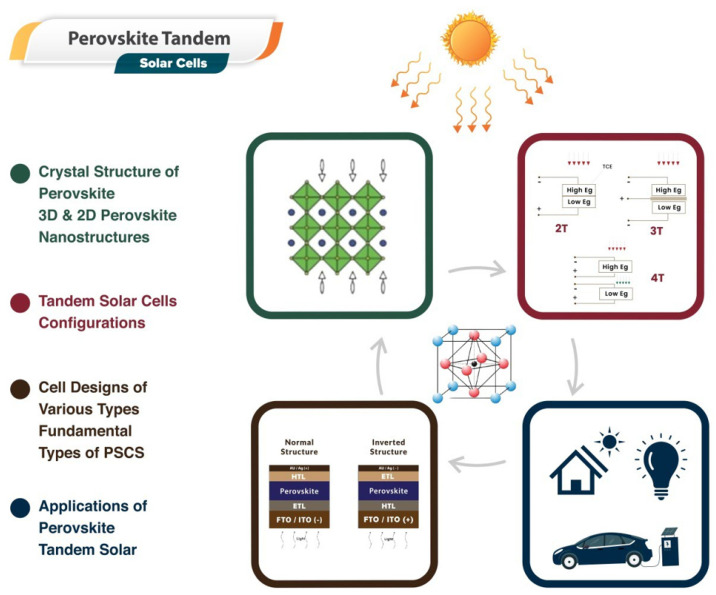
A sketch map of the tandem solar cells fabrication for industrial application.

**Figure 2 nanomaterials-13-01886-f002:**
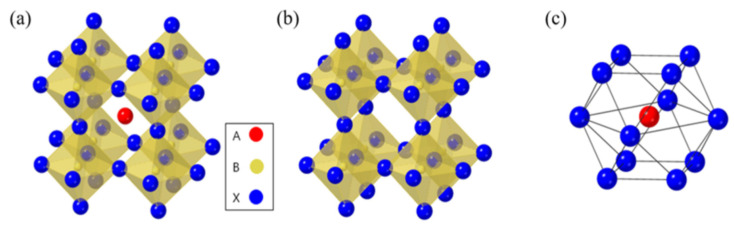
(**a**) The standard crystal structure of perovskite. (**b**) B cation and X anion crystal structure. (**c**) The A cation occupies the octahedron location of the X anion in the crystal structure [32]. Copyright 2019, MDPI.

**Figure 3 nanomaterials-13-01886-f003:**
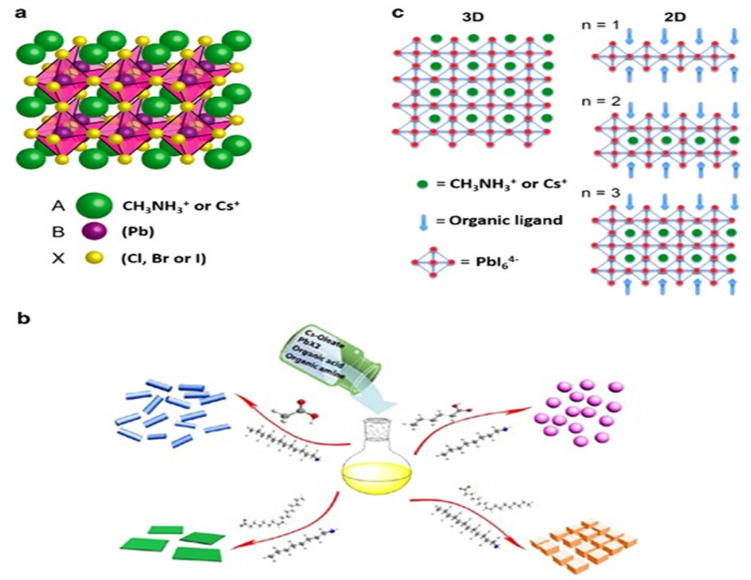
(**a**) An illustration demonstrating the crystal structure of perovskite [39]. Copyright 2015, American Chemical Society. (**b**) Adjusting the organic ligands used to tune the dimensions of inorganic perovskites [41]. Copyright 2016, American Chemical Society. (**c**) Schematic depiction of stacked 3D and (quasi)-2D perovskite nanostructures [42]. Copyright 2015, American Chemical Society.

**Figure 4 nanomaterials-13-01886-f004:**
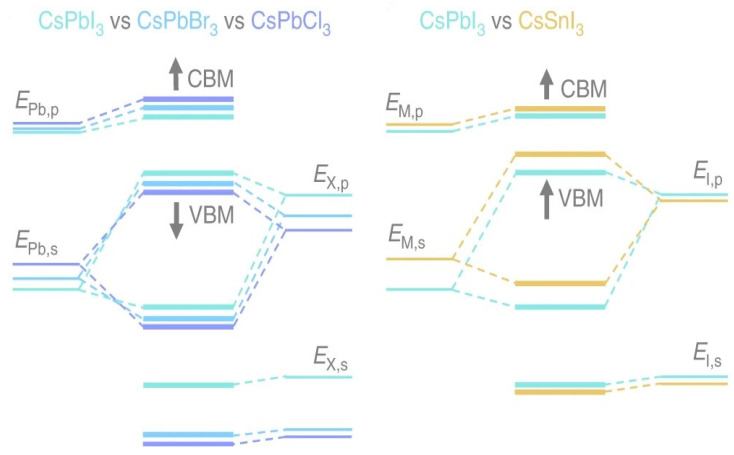
Different compositions of 3D halide perovskites have different energy levels [48]. Copyright 2019, Nature.

**Figure 5 nanomaterials-13-01886-f005:**
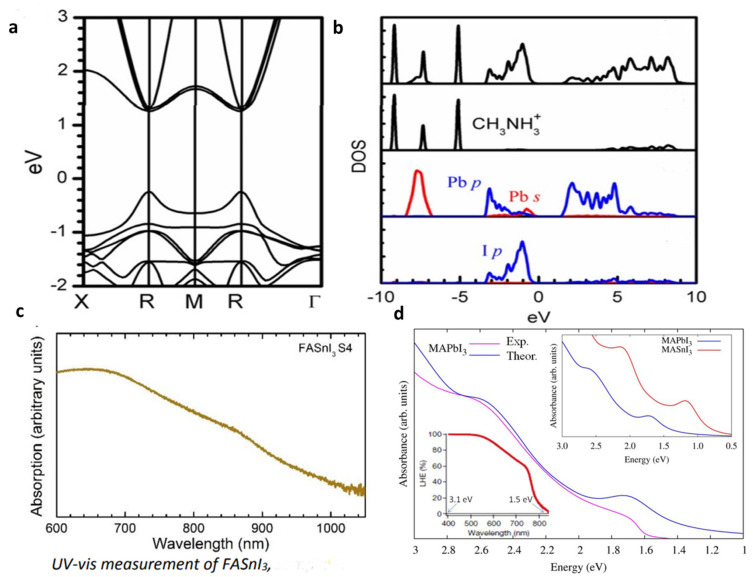
The CH_3_NH_3_PbI_3_ electronic structure. (**a**) The band structure and (**b**) calculations of the DOS for CH_3_NH_3_PbI_3_ utilizing DFT–PBE [49]. Copyright 2014, AIP Publishing. (**c**) UV–vis measurement of FASnI_3_ absorption, [48]. Copyright 2019, Springer Nature. (**d**) Comparison between the experimental UV–vis spectrum of MAPbI_3_ (red line) and the SOC–GW calculated spectrum (blue line) [51]. Copyright 2014, PubMed.

**Figure 6 nanomaterials-13-01886-f006:**
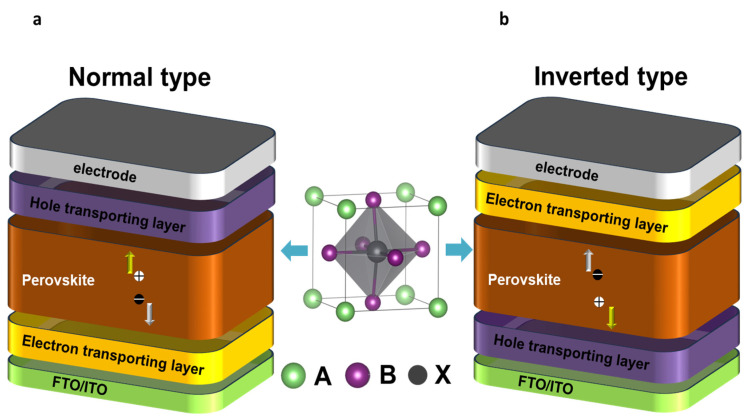
PSCS devices may be divided into two fundamental types. (**a**) Normal Type structure of PSC (**b**) Inverted type structure of PSC.

**Figure 8 nanomaterials-13-01886-f008:**
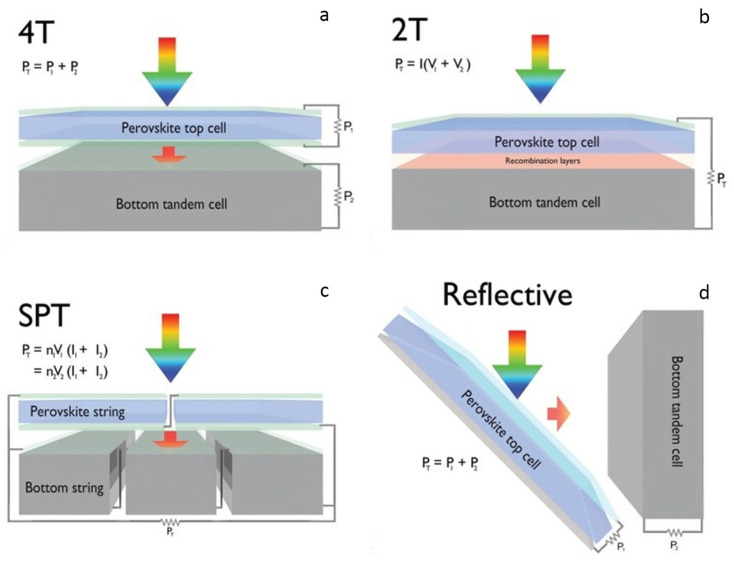
Different degrees of electrical and optical independence in tandem configurations. (**a**) The 4T tandem structure has autonomous electrical connectivity to its two cells. (**b**) A tandem comprising 2T series-connected elements. (**c**) Series-parallel tandem, voltage-matched sequences of top and bottom cells. (**d**) A reflective tandem configuration infrared (IR) reflector is positioned on the angled high-band gap cell [103]. Copyright 2021, Elsevier.

**Figure 9 nanomaterials-13-01886-f009:**
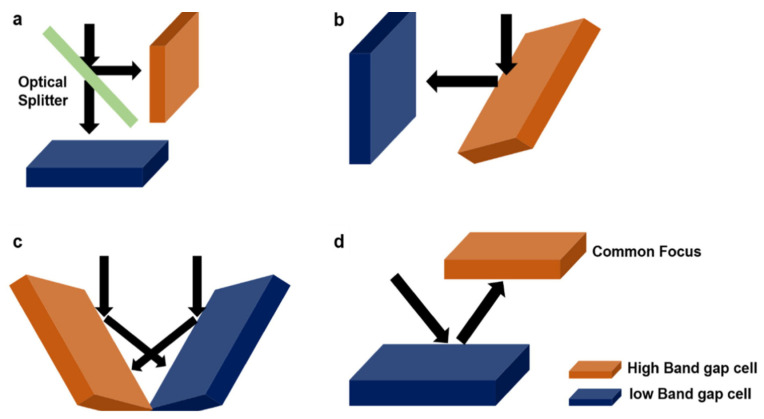
(**a**–**d**) A tandem for spectrum splitting has been proposed, which utilizes an optical splitter with a reflective tandem [109]. Copyright 2021, Elsevier.

**Figure 10 nanomaterials-13-01886-f010:**
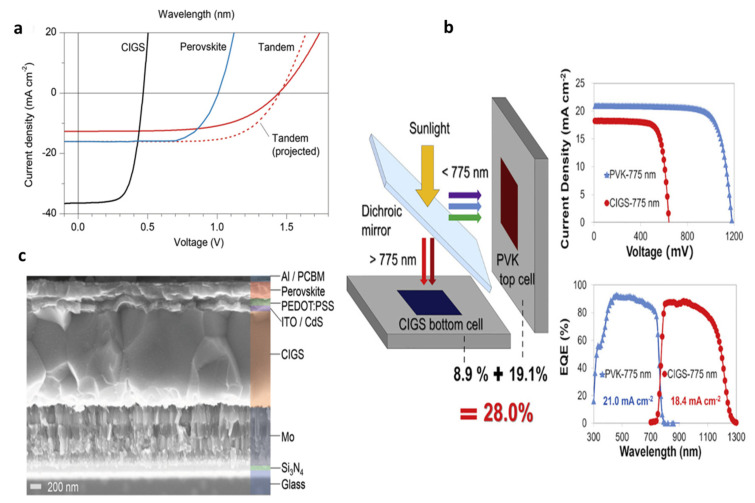
(**a**) J–V curves of the gaining tandem cell and the comparable perovskite and CIGS cells for standalone use [154]. Copyright 2015, John Wiley and Sons. (**b**) A dichroic mirror-based spectrum splitting system [156]. Copyright 2020, Elsevier. (**c**) SEM image of the structure of the Al/PCBM/perovskite/PEDOT:PSS/ITO/CdS/CIGS/Mo/Si3N4/glass TSCs [154].

**Figure 11 nanomaterials-13-01886-f011:**
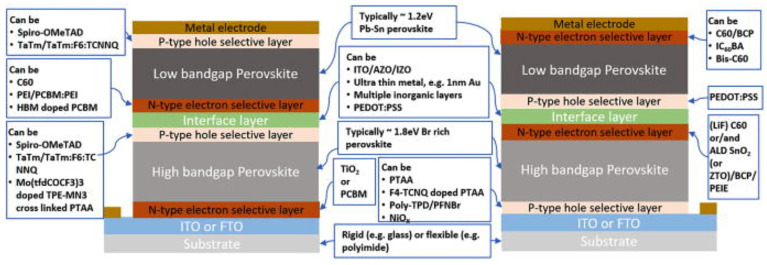
Cell designs of various types and the issues that come with them [8]. Copyright 2021, AIP Publishing. Copyright 2015, John Wiley and Sons.

**Figure 12 nanomaterials-13-01886-f012:**
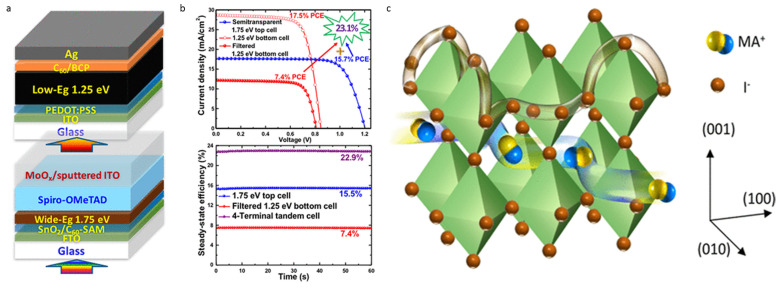
(**a**) All perovskite tandem solar cells with a 1.25 eV (FASnI_3_)_0.6_(MAPbI_3_)_0.4_ top cell: schematic illustration [177]. Copyright 2018, American Chemical Society. (**b**) J–V curves and steady-state efficiencies [177]. Copyright 2018, American Chemical Society. (**c**) Illustration of ion migration in the perovskite lattice [215]. Copyright 2016, American Chemical Society.

**Table 1 nanomaterials-13-01886-t001:** Comparison of perovskite tandem solar cells.

Device Structure	Tandem Type	Top/ Bottom	V_oc_ [V]	J_sc_ [mAcm^−2^]	FF [%]	PCE [%]	Ref
ITO/PTAA/MAPbI_3_/PCBM/C60/BCP/Cu/Au	4T	Top	1.08	20.6	74.1	16.5	[57]
IZrO/SnO_2_/Cs_0.05_FA_0.81_MA_0.14_PbI_2.55_ Br_0.45_ /Spiro/MoO_3_/IZO/MgF_2_	4T	Top	1.12	22.3	77.7	19.4	[116]
ITO/PEDOT:PSS/MA_0.5_FA_0.5_Pb_0.75_Sn_0.25_ I_3_/PCBM/Bis-C60/Ag	4T	Bottom	0.76	9.14	80	5.56	[67]
FTO/SnO_2_/C60SAM/FA_0.3_MA_0.7_PbI_3_/ Spiro-OMeTAD/MoO*_x_*/Au/MoO_x_	4T	Top	1.141	20.1	80	18.3	[166]
FTO/bl-TiO_2_/MAPbBr_3_/PTAA/PCBM/ MAPbI_3_/PEDOT:PSS/ITO	2T	Top	2.25	8.3	56	10.4	[125]
ITO/SnO_2_/C60-SAM/SiO_2_-NP/(FA_0.83_MA_0.17_)_0.95_Pb (I_0.83_Br_0.17_)_3_/Spiro-OMeTAD/MoO_3_	4T	Bottom	1.18	18.6	67.6	15.0	[140]
FTO/c-TiO_2_/m- TiO_2_/PbI_2_/CH_3_NH_3_I/Spiro-MeOTAD/ MoOx/Ag	4T	Top	0.93	18.5	51.9	11.6	[77]
FTO/c-TiO_2_/m-TiO_2_/perovskite/SpiroOMeTAD/Au	4T	Top	1.11	23.6	74	19.4	[134]
ITO/(PFN-Br)/perovskite/LiF/C_60_/(BCP)/Ag	2T	Top	1.886	19.12	75.3	18.53	[167]
ITO/TiO_2_/perovskite/Sproro-OMeTAD/MoO_3_/Au	4T	Top	1.156	19.8	79.9	18.3	[78]
ITO/PTAA/(FA_0.65_MA_0.20_Cs_0.15_)Pb(I_0.8_Br_0.2_)_3_/C60/Ag	2T	Top	0.677	35.11	76	17.28	[144]

**Table 2 nanomaterials-13-01886-t002:** A comparison of 2T and 4T tandem solar cells.

2T	4T
It consists of two series connected cells of different bandgaps, requiring current matching. It is not easy to fabricate. 2.1. Fabrication of recombination layers with a minimal loss between the cells. 2.2. Optical management within the cells. The top cell is directly integrated with the bottom cell. Optical coupling is required. It requires fewer processing steps and fewer contact layers. Less parasitic absorption from the glass substrate. Lifetime of the tandem cell will be determined by the perovskite top cell.	These cells are not necessarily connected in series; these are not limited by current matching. It is comparatively easy to fabricate. 2.1. All the terminals of the sub cells are operated individually to get maximum power. 2.2. This four terminal device works in such a way that two diodes mean the incident light is split into two diodes, and these diodes work electrically independently. Fabricated independently. Optical coupling is required. Easy to maintain and fix by replacing new subcell. The overall device efficiency is not sensitive to the solar spectrum. Conductive glass substrate of the perovskite top cell will cause optical losses due to its parasitic absorption.

## Data Availability

Not applicable.

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
