# Peer review of "Recent Progress in Perovskite Tandem Solar Cells"

_nanomaterials, 2023, doi:10.3390/nano13121886_

Round 1
Reviewer 1 Report
In this manuscript, the authors have reviewed the recent advancements, device configurations and current challenges in perovskite-based TSCs and explored the ways to boost the power conversion efficiencies and the stability of devices. This work is meaningful and well organized, which can be accepted for publication in Nanomaterials after a minor revision by addressing the following comments.
1 A brief discussion about the perovskite subcells with different fundamental types in TSCs is required.
2 The advantages and disadvantages of different bottom cells in perovskite-based TSCs should be discussed.
3 The subtitles in 2T perovskite tandem solar cells should be revised.
4 Some new and relevant articles should be cited such as Journal of Energy Chemistry 83 (2023) 189-208; Energy Fuels 37 (2023) 6401-6423.
Author Response
Reviewer 1
The authors give sincere thanks to Reviewer for valuable comments and recommendations which let us improve substantially our paper. Please find below answers to the questions and remarks.
- A brief discussion about the perovskite subcells with different fundamental types in TSCs is required.
- The advantages and disadvantages of different bottom cells in perovskite-based TSCs should be discussed.
- The subtitles in 2T perovskite tandem solar cells should be revised.
- Some new and relevant articles, such as Journal of Energy Chemistry 83 (2023) 189-208; Energy Fuels 37 (2023) 6401-6423, should be cited.
Answers
- A detailed discussion is already provided with different fundamental types in each section, such as Wideband gap for perovskite cell, Narrowband gap for bottom cell, 4T tandem solar cells, Four-Terminal CIGS-Based Solar Cells, 4T Perovskite/Si Tandem Solar Cells, etc. are highlighted.
- Two tables have been added to address the challenges and characteristics of different tandem solar cells. (Table 1, Table 2)
- The subtitle in 2T perovskite tandem solar cells has been revised. The figure in that part is also provided in two parts named Figure 10 and Figure 11 for better understanding.
- The suggested articles are cited.

Reviewer 2 Report
The submission is interesting because it provides an overview of tandem technology for perovskite cells. The article is extensive and well written enough to be considered for publication in Nanomaterialas. However, some revision is required.
Formatting is incorrect -- writing equations is wrong - please correct. Overall, the scientific insight into the functionality of perovskite cells is poor and ranged in the article to only a phenomenology\technique description of various practical solutions without going into details of the physical mechanisms. In view of the most important goal of the tandem concept of photovoltaic cells, which is to increase the efficiency of complex cells, this shortcoming in the presentation should be reduced. At least a more detailed description/discussion of the Shockley–Queisser limit should be placed with the emphasis that this limit was originally calculated for p-n junction cells, and not for hybrid-chemical cells like perovskite cells. The difference in the way the two types of cells operate should be explained and some important points determining/increasing the efficiency of perovskite cells should be pointed out (cf. Laska et al. Nano Energy 75 (2020) 104751). In this connection, a very promising method of improving perovskite tandem cell efficiency should be mentioned in order to better adapt the structure to the spectrum of solar radiation in the tandem cell solution -- it can be influenced not only by the chemical structure of the materials used (mainly related to the forbidden gap in tandem semiconductors), but through additional plasmonic nanocomponents or quantum dots (especially easy to apply to liquid perovskite precursors) (Materials 2022, 15, 2254; J. Phys. Chem. C 2016, 120, 13, 6996-7004). Discussing the reasons for the limitations of solar cell efficiency (including tandems) would increase the scientific value of the presentation. In reviewing tandem solutions, more complete literature should be considered – e.g., an important work should be mentioned (Rajagopal et al. Adv. Mater. 2017, 29, 1702140). Bibliography should be corrected (in [87} the journal name is avoided).
Next, please avoid acronyms in the abstract and move them to the Introduction. Correction of the text is suggested (e.g. citations of references should be separated from the text, evenly throughout the paper). Also, a more informative commentary on the current status of perovskite cells would be useful at the beginning of the Introduction. The sentence in the introduction "The devices' efficiency is close to the theoretical maximum limit of 29.4%[4,5]. " needs to be explained because the theoretical limit is larger (cf. Shockley–Queisser limit). Linguistic polishing is recommended. The Conclusions should be corrected to make the comments more precise and targeted.
linguistic polishing is required
Author Response
Reviewer 2
The authors give sincere thanks to Reviewer for valuable comments and recommendations which let us improve substantially our paper. Please find below answers to the questions and remarks.
1) Formatting is incorrect—writing equation is wrong-please correct.
Answer. Goldschmidt tolerance factor t (equation (1) of manuscript) is used to predict the formability of different kinds of perovskite. The rule was developed for oxide perovskite, but the trend is still valid for inorganic-organic hybrid halide materials (Li et al. Chem.Mater. 2016, 28,284-292). Goldschmidt tolerance factor t is the zeroth-order approximatiease that facilates ease of calculation in most of the cases, but anomaly has been found in approximately 26%of cases ( Dunfield et al. Adv.Energy Mater. 2020, 1904054; Mazumdar et al. Frontiers in Electronics 2021, V.2, 712785). Due to this, Bartel et al. (Sci.Adv. 2019,5,eaav0693) proposed a modified tolerance factor t , which can predict the probability of perovskite formation more accuratelly than that of equation (1), but modified tolerance factor t cannot predict the exact crystallografic structures such as equation (1) (Dunfield et al. Adv.Energy Mater. 2020, 1904054; Mazumdar et al. Frontiers in Electronics 2021, V.2, 712785). Therefore, for the analysis of perovskite structures, we used equation (1).
2) At least a more detailed description/discussion of the Shockley–Queisser limit should be placed with the emphasis that this limit was originally calculated for p-n junction cells, and not for hybrid-chemical cells like perovskite cells.
Answer. In introduction we added: The devices' efficiency is close to the theoretical maximum limit of 33.3% according to Shockley-Queisser's (SQ) detailed-balance model for ideal p-n junction [4]. A key limiting factor not accounted for in the SQ model is the Auger recombination of free carriers that occurs under illumination. Taking this into account for silicon, the efficiency limit for monocrystalline Si SC with an optimized thickness (110 mm) was calculated to be 29.4% [5,6].
3) The difference in the way the two types of cells operate should be explained and some important points determining/increasing the efficiency of perovskite cells should be pointed out (cf. Laska et al. Nano Energy 75 (2020) 104751). In this connection, a very promising method of improving perovskite tandem cell efficiency should be mentioned in order to better adapt the structure to the spectrum of solar radiation in the tandem cell solution -- it can be influenced not only by the chemical structure of the materials used (mainly related to the forbidden gap in tandem semiconductors), but through additional plasmonic nanocomponents or quantum dots (especially easy to apply to liquid perovskite precursors) (Materials 2022, 15, 2254; J. Phys. Chem. C 2016, 120, 13, 6996-7004).
Answer. In introduction we added: The concept of intermediate-band solar cells has been proposed to achieve a conversion efficiency of 63% [7]. Quantum dots have been considered one of few materials systems to form intermediate bands for intermediate-band solar cells [8]. Process of carrier multiplication in which a single photon generated two (or more) electro-hole pairs can enhance the photocurrent of SCs [9]. Metallization at the nanoscale is also one method to improve SCs' efficiency [10].
In chapter 4 we added; The application of metallic nanoparticles leads to increased efficiency of SCs due to the plasmonic effect [10,115]. This became possible by using light trapping through the resonant scattering and concentration of light in arrays of metal nanoparticles or by coupling light into surfaces plasmon polaritons and photonic modes that propagate in the plane of the semiconductor layer. Metal nanoparticles can excite localized surface plasmon resonance, leading to increased light absorption and increasing the efficiency of SCs [116]. A significant photocurrent increase induced by metallic nanoparticles was observed in perovskite SC with incorporated Au/SiO2 core-shell nanoparticles [117]. When silica-coated gold (Au@SiO2) nanorods are embedded in the interface between the hole transport layer PEDOT: PSS and the perovskite CH3NH3PbI3, the average PCE increased over 40% from 10.9% for PSCs without Au@SiO2 to 15.6% with Au@SiO2 [118]. According to Jack et al., this enhancement can be due to the reduction of the binding energy of excitons by plasmons, which eventually accelerates the dissociation of excitons at the interface with the electron transport layer [10].
4) In reviewing tandem solutions, more complete literature should be considered – e.g., an important work should be mentioned (Rajagopal et al. Adv. Mater. 2017, 29, 1702140).
Answer. At chapter 4.2.2 we added: A highly efficient 2T tandem SC reaching 80% of the theoretical limit in voltage was developed by Rajagopal et al. [184]. Optical simulations were utilized to validate the current matching criterion and identify possibilities for further improving the current generation in tandem architecture. Pb-Sn binary perovskite (MAPb0.5 Sn0.5I3) with a low bandgap of 1.2 eV was used as the bottom subcell, and WB perovskite Cs0.1MA0.9Pb(I0.6Br0.4)3 with a bandgap of 1.8 eV served as a top cell. Current-voltage characteristics of the best-performing 2T TSC show an exceptional Voc of 1.98 V, a Jsc of 12.7mA cm-2, and an FF of 0.73, resulting in a PCE of 18.4%.
5) Bibliography should be corrected (in [87} the journal name is avoided). Next, please avoid acronyms in the abstract and move them to the Introduction. Correction of the text is suggested (e.g. citations of references should be separated from the text, evenly throughout the paper). The Conclusions should be corrected to make the comments more precise and targeted.
Answer. Journal name is added. Acronyms in abstract are removed. Citations of references are improved. The conclusions we try to improve.
6) The sentence in the introduction "The devices' efficiency is close to the theoretical maximum limit of 29.4%[4,5]. " needs to be explained because the theoretical limit is larger (cf. Shockley–Queisser limit).
Answer. In introduction the sentence was partially changed.

Reviewer 3 Report
Ašmontas and Mujahid report on the recent progress in perovskite tandem solar cells. The manuscript constitutes an excellent contribution to a hot field and thus, it is my pleasure to recommend it for publication in nanomaterials.
Moreover, I would like to point out the following:
1) Page 2, line 63. Please add the oxidation state of the metal to CH3NH3PbI3:
CH3NH3PbIII3. Please do this throughout the whole manuscript.
2) Page 3, line 116. Please change Pb2+ to PbII. Please do this throughout the whole manuscript and for all the metals.
3) Figures 3 and 4 are blurring.
4) Figure 9 contains so much information and thus, it is difficult to read it.
Author Response
Reviewer 3
The authors give sincere thanks to Reviewer for valuable comments and recommendations which let us improve substantially our paper. Please find below answers to the questions and remarks.
Questions.
- Page 2, line 63. Please add the oxidation state of the metal to CH3NH3PbI3: CH3NH3PbIII3. Please do this throughout the whole manuscript.
- Page 3, line 116. Please change Pb2+ to PbII. Please do this throughout the whole manuscript and for all the metals.
- Figures 3 and 4 are blurring.
- Figure 9 contains so much information; thus, it isn't easy to read.
Answers
- We have checked from literature everyone is using the same formula, CH3NH3PbI3: instead of CH3NH3PbIII3. Some see the provided references. https://www.nature.com/articles/s41598-020-68085-0, https://pubs.acs.org/doi/10.1021/acsnano.5b07791, https://www.science.org/doi/abs/10.1126/science.1243167,
- We have checked from literature everyone is using the same formula, Pb2+: instead of PbII. Some see the provided references. https://link.springer.com/article/10.1007/s10854-023-10318-9https://link.springer.com/article/10.1007/s10854-023-10004-whttps://chemistry-europe.onlinelibrary.wiley.com/doi/abs/10.1002/slct.202300520
- Figure 3,4 has been updated
- We divided it into two parts, Figure 9 and Figure 10.

Reviewer 4 Report
The authors Asmontas et al. reviewed the recent progress in Perovskite Tandem Solar Cells including material characterization, device design and working mechanisms. This manuscript is interesting and well-shaped. However, some minor issues are better to be addressed before accepted.
1. A short paragraph should be added to the introduction to give an idea of the outline of the manuscript. A sketch map is required to be added to describe the relationship between each part.
2. The first sentence of each part should be changed from first-person active voice to passive voice. E.g. line 119, “we see that the molecular structure……”, line 192, “we’ll take a high-level……”.
3. For part 2.2, some experimental evidence of typical perovskite materials from literature is better to be provided as well to convince the modelling results. E.g. UPS, UV-Vis adsorption, Tauc plot, etc.
4. A table to summarize the activity, advantages and disadvantages of each device in part four is suggested to be added.
5. Critical brief summaries are better to be added to each part.
Sentences in the first-person active voice are needed to be changed to the passive voice. E.g. line 119, “we see that the molecular structure……”, line 192, “we’ll take a high-level……”.
Author Response
Reviewer 4
The authors give sincere thanks to Reviewer for valuable comments and recommendations which let us improve substantially our paper. Please find below answers to the questions and remarks.
Questions
- A short paragraph should be added to the introduction to give an idea of the outline of the manuscript. A sketch map must be added to describe the relationship between each part.
- The first sentence of each part should be changed from first-person active voice to passive voice. E.g., line 119, “we see that the molecular structure……”, line 192, “we’ll take a high-level……”.
- For part 2.2, some experimental evidence of typical perovskite materials from the literature is better to be provided to convince the modeling results—E.g., UPS, UV-Vis adsorption, Tauc plot, etc.
- A table to summarize the activity, advantages, and disadvantages of each device in part four is suggested to be added.
- Critical summaries are better to be added to each part.
Answers
- A short paragraph and sketch (Figure 1) are provided
- The sentence structure has been changed.
- We have added more detail in Figure 4.
- Two tables have been added to address the challenges and characteristics of different tandem solar cells. (Table 1, Table 2)
- Critical summaries are highlighted.

Round 2
Reviewer 2 Report
Authors have corrected the submisson and responded to all former recommendations in a satisfactory way; the paper can be published as a review of an interesting topic. Final proofreading is, however, required -- cf . e.g., line 1219 ( the capital letter is not necessary and a discovery can be done only one time)
a final minor proofreading is required
Author Response
Dear Reviewer,
I would like to say special thank to you for your time and support. We have corrected the highlighted problem by you specifically in the line number 1219. We have carefully check the whole article again and it is correct version.
